# Methylation of ESCRT-III components regulates the timing of cytokinetic abscission

Aurélie Richard[1,7], Jérémy Berthelet[1,7], Delphine Judith[2], Tamara Advedissian [3], Javier Espadas [4], Guillaume Jannot[1], Angélique Amo [1], Damarys Loew [5], Berangere Lombard [5], Alexandre G. Casanova[6], Nicolas Reynoird [6], Aurélien Roux [4], Clarisse Berlioz-Torrent[2], Arnaud Echard [3], Jonathan B. Weitzman[1] & Souhila Medjkane [1] ✉

Abscission is the final stage of cytokinesis, which cleaves the intercellular bridge (ICB) connecting two daughter cells. Abscission requires tight control of the recruitment and polymerization of the Endosomal Protein Complex Required for Transport-III (ESCRT-III) components. We explore the role of post-translational modifications in regulating ESCRT dynamics. We discover that SMYD2 methylates the lysine 6 residue of human CHMP2B, a key ESCRT-III component, at the ICB, impacting the dynamic relocation of CHMP2B to sites of abscission. SMYD2 loss-of-function (genetically or pharmacologically) causes CHMP2B hypomethylation, delayed CHMP2B polymerization and delayed abscission. This is phenocopied by CHMP2B lysine 6 mutants that cannot be methylated. Conversely, SMYD2 gain-of-function causes CHMP2B hypermethylation and accelerated abscission, specifically in cells undergoing cytokinetic challenges, thereby bypassing the abscission checkpoint. Additional experiments highlight the importance of CHMP2B methylation beyond cytokinesis, namely during ESCRT-III-mediated HIV-1 budding. We propose that lysine methylation signaling fine-tunes the ESCRT-III machinery to regulate the timing of cytokinetic abscission and other ESCRT-III dependent functions.

The ESCRT (Endosomal Sorting Complex Required for Transport) machinery is evolutionarily conserved and orchestrates diverse cellular processes that require membrane remodeling and membrane fission[1–3]. Four distinct ESCRT complexes (ESCRT-0, I, II, and III) play crucial roles in a range of processes[1], such as formation of intraluminal vesicles, reformation of the nuclear envelope, plasma membrane, organelle repair, autophagy, virus budding and release of viral particles. Notably, ESCRT-III proteins are required for the abscission process during cytokinesis in mammals, as well as in Archaea, supporting an ancestral role of the ESCRT pathway in cell division[4]. Cytokinesis is

the final stage of cell division and consists of a rapid step of furrow ingression followed by the abscission process, which lasts for several hours in mammalian cells, until the ESCRT machinery splits the membrane apart[5–7]. Abscission leads to the physical separation of the two daughter cells through the scission of the intercellular bridge (ICB) which serves as a platform to assemble the abscission machinery[8,9]. Reconstitution assays in vitro showed that yeast *S. cerevisiae* Snf7 (CHMP4 ortholog) recruits Vps2-Vps24 (CHMP2-CHMP3 orthologs) followed by Did2 and Ist1 (CHMP1 and Ist1 orthologs, respectively)[10,11]. The AAA+ ATPase Vps4 promotes the turnover of ESCRT-III filaments

[1]Université Paris Cité, CNRS, UMR7126 Epigenetics and Cell Fate, F-75013 Paris, France. [2]Université Paris Cité, Inserm, CNRS, Institut Cochin, F-75014 Paris, France. [3]Institut Pasteur, Université Paris Cité, CNRS UMR3691, Membrane Traffic and Cell Division Unit, 25-28 Rue du Dr Roux, F-75015 Paris, France. [4]Department of Biochemistry, University of Geneva, CH-1211 Geneva, Switzerland. [5]Institut Curie, PSL Research University, Centre de Recherche, CurieCoreTech Mass Spectrometry Proteomics, F-75005 Paris, France. [6]Université Grenoble Alpes, CNRS UMR5309, INSERM U1209, Institute for Advanced Biosciences, 38000 Grenoble, France. [7]These authors contributed equally: Aurélie Richard, Jérémy Berthelet. ✉e-mail: souhila.medjkane@u-paris.fr

and subunit exchange to catalyze membrane fission[12,13]. At the early stage of abscission in human cells, the ESCRT-III members (including CHMP2B) are recruited to form two ring structures surrounding the midbody, at the center of the ICB[12,14]. Subsequently, ESCRT-III proteins polymerize to form hetero-polymer, helical spiral structures with progressively smaller diameters. This leads to membrane constriction at the abscission site, located approximately 1 μm from the midbody, thereby promoting abscission[11,15]. The precise mechanisms controlling the spatio-temporal dynamics of ESCRT-III components and the timing of abscission are not fully understood. Specifically, the mechanisms regulating the polymerization of ESCRT-III proteins from the initial assembly site at the midbody (rings) to the abscission site remain unclear. Protein interactions and post-translational modifications (PTMs) may facilitate ESCRT-III-mediated scission across various spatial and temporal scales. For example, phosphorylation of CHMP4C by Aurora B kinase delays abscission in response to abscission checkpoint activation[16,17], and ubiquitination of ESCRT-III is critical to complete abscission in *Drosophila*[18]. The roles of other PTMs, such as methylation, in ESCRT regulation has not been investigated.

Lysine methylation is a key PTM, mostly studied in the context of histone methylation and epigenetic regulation of gene expression. However, non-histone protein methylation is emerging as a regulator of broader cellular processes[19,20], such as cell proliferation and differentiation[21,22]. The importance of methylation in regulating non-histone protein function in physiological and disease contexts is attracting much attention, particularly as the enzymes involved are emerging as promising drug tartgets[23]. SMYD2 (SET and MYND domain-containing protein 2) is one of the lysine methyltransferases that catalyzes the lysine methylation of both histone and non-histone targets[24]. SMYD2 contributes to tumor formation in a wide spectrum of cancers[25] and mice models have demonstrated a pro-oncogenic role of SMYD2 in vivo[26–30]. Despite considerable efforts, little is known about the physiological and cellular roles of SMYD2 and how its over-expression contributes to the development of a wide range of tumors.

Here, we report the biochemical and functional characterization of a new PTM on a key ESCRT-III member and its impact during cytokinesis. We discovered that SMYD2 methylates the CHMP2B lysine 6 residue (CHMP2B K6) at the ICB of cytokinetic cells. Methylation of CHMP2B controls the distribution of ESCRT-III polymers at the ICB during cytokinesis and accelerates the abscission in cells undergoing mitotic stress or abscission checkpoint activation. Our study reveals that CHMP2B K6 plays a pivotal role in both abscission and HIV viral budding, highlighting the importance of this methylation event. These findings also suggest that lysine methylation signaling could trigger premature exit from the abscission checkpoint in cancer cells that overexpress SMYD2, and contribute to oncogenesis. Methylation appears as a fine-tuning timing mechanism for key ESCRT-III cellular processes. Our findings provide insight into the significance of methylation in ESCRT biology.

## Results
### SMYD2 localizes at the midbody and methylates the CHMP2B protein
The SMYD2 enzymes was first described as a histone methyltransferase, but subsequent literature suggests that its natural substrates are non-histone proteins. To investigate the cellular functions of the SMYD2 lysine methyltransferase, we monitored the localization of GFP-tagged SMYD2 in HeLa cells using live-cell imaging. We observed that GFP_SMYD2 primarily localized to the cytoplasm, with a portion of SMYD2 recruited to the intercellular bridge during cytokinesis (Fig. 1A and Supplementary Movie 1). This localization pattern raised the potential involvement of SMYD2 in cytokinesis by methylation of midbody proteins. To identify potential substrates of SMYD2 at the midbody, we explored two proteomic datasets: (1) data from a methyl-lysine proteomic screen to identify SMYD2-dependent methylation sites in

KYSE-150 cells[31], and (2) a Flemmingsome dataset from a midbody remnant proteome of 489 proteins enriched in post-abscission midbodies from HeLa cells[32]. By comparing these two datasets, we identified 16 common proteins, three of which are known to be involved in cytokinesis: the kinesin KIF14[33], the p50RhoGAP[34], and the ESCRT-III component CHMP2B[35] (Fig. 1B). We focused on the CHMP2B ESCRT-III protein as a potential SMYD2 target for further studies. To explore the potential interaction between SMYD2 and CHMP2B, we performed co-immunoprecipitation experiments in HeLa cells, which revealed the formation of a complex between these two proteins (Fig. 1C). Subsequently, we tested whether SMYD2 could directly methylate CHMP2B using in vitro H3-radiolabeled S-adenosyl-methionine (SAM) methylation assays. We demonstrated the direct methylation of CHMP2B by recombinant SMYD2 enzyme, but not by the catalytic-dead (CD) mutant SMYD2 Y240A (Fig. 1D). To identify the specific methylated lysine residue in CHMP2B, we performed mass spectrometry (MS)-based proteomic analysis of endogenous CHMP2B in HeLa cells. We detected mono-methylation at lysine 6 of endogenous CHMP2B (CHMP2B K6) in HeLa cell extracts (Fig. 1E). To quantify the level of methylated CHMP2B K6, we performed label-free quantification using CHMP2B transiently transfected with either Flag-tagged SMYD2 or the catalytic-dead mutant SMYD2 Y240A, followed by MS analysis of immunoprecipitated CHMP2B_GFP. The results showed a significant increase in K6 mono-methylation (K6me1) upon expression of SMYD2 compared to the SMYD2 Y240A mutant (Fig. 1F), demonstrating that SMYD2 catalytic activity is required for mono-methylation of CHMP2B K6 in HeLa cells. We conducted mutagenesis experiments to confirm that CHMP2B K6 is the main lysine target of SMYD2. In vitro methylation assay using a lysine-to-alanine CHMP2B K6A recombinant mutant protein as substrate, revealed the abrogation of CHMP2B methylation, thus confirming the direct and exclusive methylation of CHMP2B K6 by SMYD2 (Fig. 1G and Supplementary Fig. 1A). This result was further validated using GFP_CHMP2B and the CHMP2B K6A mutant, transiently transfected and immunoprecipitated from HeLa cells (Supplementary Fig. 1A). Furthermore, we used fluorescently-coupled CHMP2B peptides containing K6 to demonstrate efficient in vitro methylation of the unmodified peptide by SMYD2. In contrast, the K6me1 peptide could not undergo further methylation (Supplementary Fig. 1B). This indicates that SMYD2 specifically mono-methylates CHMP2B and does not efficiently proceed to di- or tri-methylation. Additionally, two specific SMYD2 inhibitors, BAY-598 or LLY-507, completely abolished methylation of the CHMP2B peptide (Supplementary Fig. 1C), demonstrating the druggability of this methylation. Taken together, these biochemical results establish CHMP2B as a genuine cytoplasmic substrate of SMYD2.

### SMYD2 methylates CHMP2B K6 at the intercellular bridge of dividing cells
To evaluate the role of CHMP2B K6 methylation in cells, we first generated an antibody that can specifically recognize CHMP2B K6 me1 in vivo. The anti-CHMP2B K6 me1 antibody successfully detected CHMP2B_GFP in the presence of Flag_SMYD2, but not in the presence of the catalytic-dead Flag_SMYD2 Y240A mutant enzyme in total protein extracts (Fig. 2A). Furthermore, the CHMP2B K6 me1 antibody did not detect the K6me1 signal in protein extracts from cells expressing the GFP_CHMP2B K6A mutant, demonstrating the high specificity for K6me1 (Fig. 2A). These findings were further validated using recombinant proteins (Supplementary Fig. 2A) and the methylation signal was lost upon treatment with pharmacological inhibitors of SMYD2 (Supplementary Fig. 2A). This antibody detected endogenous CHMP2B K6me1 by Western blot analysis following CHMP2B immunoprecipitation and the signal diminished upon SMYD2 inhibition (Fig. 2B). Collectively, these results provide evidence that CHMP2B K6 methylation is specifically recognized by the CHMP2B K6 me1 antibody in protein extracts. Additionally, we observed a distinct endogenous CHMP2B K6 methylation signal at the ICB of dividing HeLa cells by

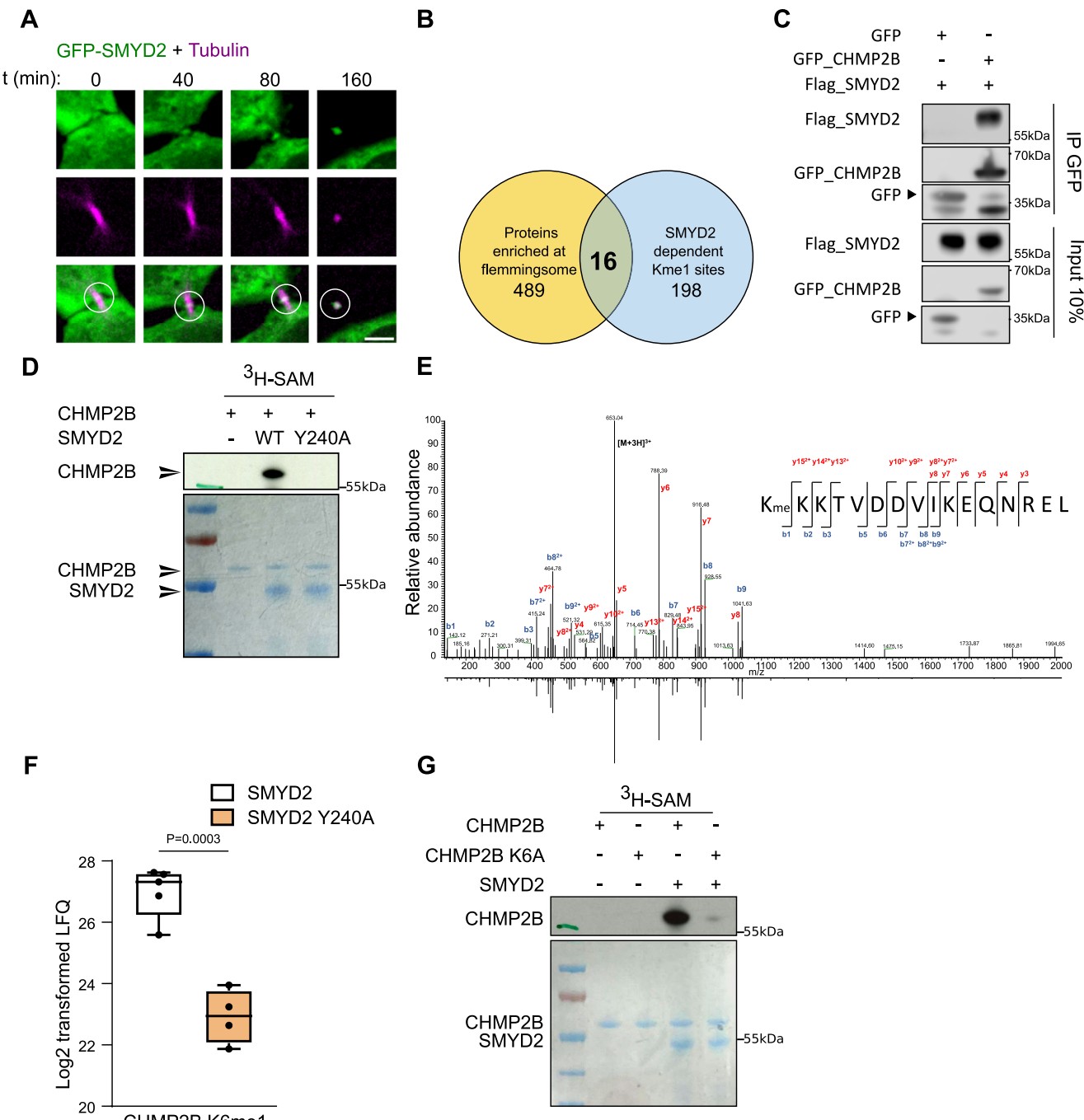

**Fig. 1 | SMYD2 localizes at the midbody and methylates the CHMP2B protein.**
**A** Representative cell images of live HeLa cells during cytokinesis, which were transiently transfected with GFP_SMYD2 and treated with siR tubulin (magenta) to stain for tubulin cytoskeleton. GFP_SMYD2 localizes to the midbody (circle). Scale bar = 5 um (*N* = 3). **B** The Venn diagram showing common proteins found between the 489 proteins identified in remnant midbodies from HeLa cells[32] and the 198 reproducible SMYD2 dependent Kme1 sites detected through SILAC analysis conducted on KYSE-150 cells[31]. **C** HeLa cells were co-transfected with the constructs as indicated on the figure. Protein extracts were immunoprecipitated using a GFP antibody and the interaction between GFP_CHMP2B and Flag_SMYD2 was assessed by western blot (*N* = 3). **D** Recombinant SMYD2 (WT or catalytic dead Y240A mutant) were incubated with recombinant CHMP2B. Samples were then subjected to SDS-PAGE, and the total proteins were visualized using Coomassie staining. Gel was later dried and methylated proteins were detected by autoradiography (16 h exposure). The experiment displayed here is representative of 3 different replicates. **E** Tandem mass spectrometry (MS/MS) analysis showing mono-methylation of K6 on endogenous CHMP2B after immunoprecipitation from HeLa cells.

Sequence of the chymotryptic endogenous mono-methylated CHMP2B peptide is displayed in inset. Spectrum of the corresponding synthetic CHMP2B peptide is shown inverted. The single- and double-charged fragment ions labeled in the spectrum represent cleavage of the peptide bond, recording sequence information from the N and C termini (b- and y-type ions, respectively). **F** Boxplot representation of log2 transformed intensities of peptide CHMP2B K6me1 after immunoprecipitation, gel purification and chymotrypsin digestion of GFP_CHMP2B co expressed with SMYD2 or SMYD2 Y240A in HeLa cells (*N* = 5 for GFP_CHMP2B co-expressed with Flag_SMYD2, and *N* = 4 for GFP_CHMP2B co-expressed with Flag_SMYD2 Y240A). The boxplot indicates the 25% (bottom), 50% (center) and 75% quartiles (top). Whiskers represent the minimum (bottom) and the maximum (top). Two-sided unpaired *t*-test. **G** Recombinant SMYD2 (WT or catalytic dead Y240A mutant) were incubated in the presence of recombinant CHMP2B (WT or K6A mutant). Samples were then subjected to SDS-PAGE, and the total proteins were visualized using Coomassie staining. Gel was later dried and methylated proteins were detected by autoradiography (16 h exposure). The experiment displayed here is representative of 3 different replicates.

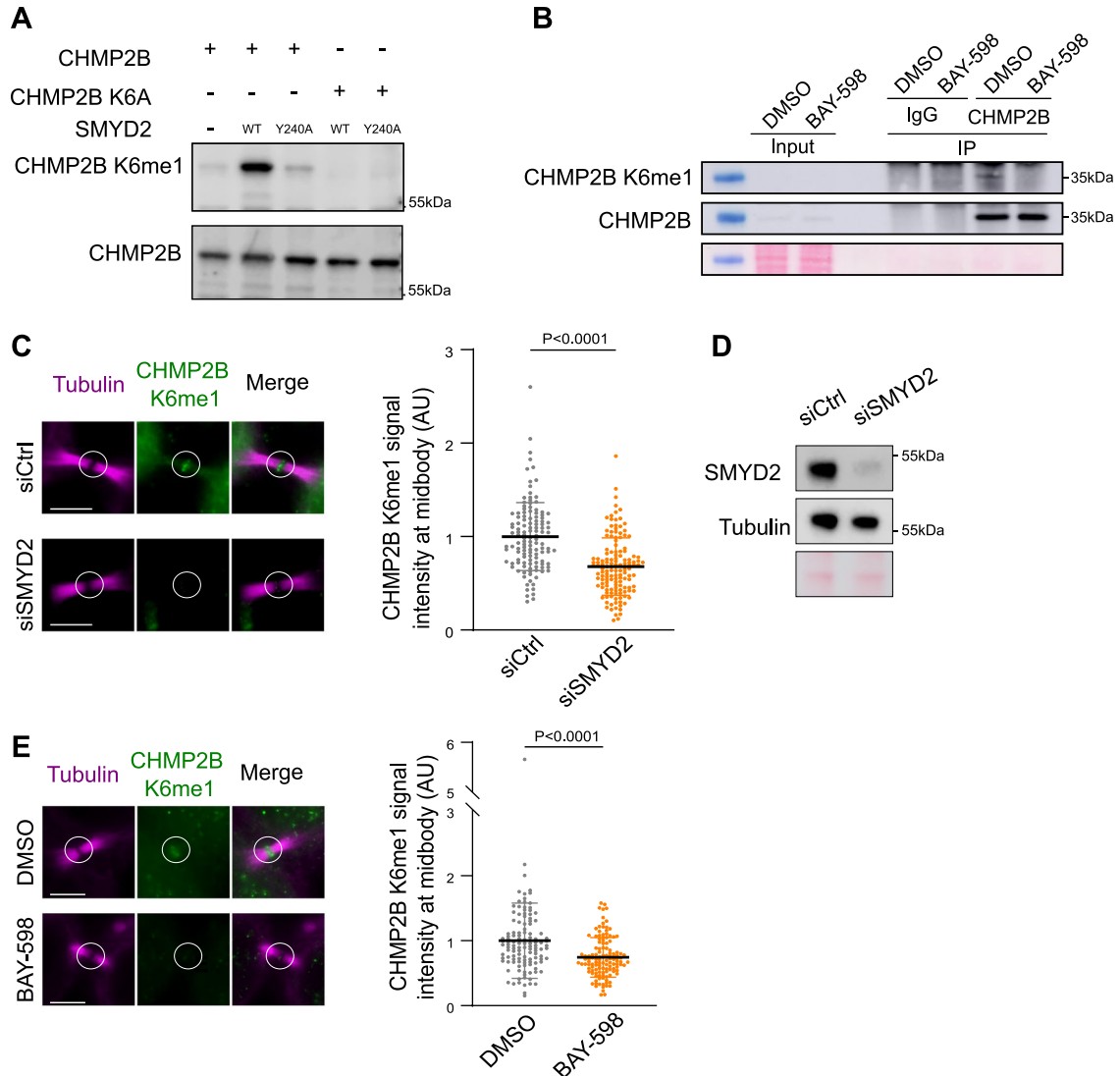

**Fig. 2 | SMYD2 methylates CHMP2B K6 at the intercellular bridge of dividing cells. A** Immunodetection of CHMP2B K6 methylation on GFP_CHMP2B using CHMP2B K6me1 antibody in HeLa cells transfected with Flag_SMYD2 or SMYD2 Y240A and GFP_CHMP2B or CHMP2B K6A mutant as indicated on the figure (*N* = 3). **B** Protein extracts from HeLa cells treated with DMSO or 10 uM of BAY-598 (SMYD2 inhibitor) were immunoprecipitated using CHMP2B antibody and the CHMP2B K6 methylation was assessed by western blot using CHMP2B K6me1 antibody (*N* = 3). **C** Left panels: Representative cell images of CHMP2B K6me1 (green) and alpha-tubulin (magenta) immunostaining in HeLa cells treated with Ctrl or SMYD2 siRNAs. The white circle shows the midbody. Scale bar = 5 um. Right panel: quantification of the CHMP2B K6me1 signal at the midbody of HeLa cells treated with Ctrl or SMYD2 siRNAs as indicated (*N* = 3, *n* = 125 for siCtrl cells and *n* = 140 siSMYD2 cells midbodies quantified by condition, mean ± SD, Two-sided Mann–Whitney test). **D** Western blots of protein extracts from HeLa treated with control (siLuciferase referred to siCtrl) or SMYD2 siRNAs, revealed with SMYD2 and tubulin antibodies (*N* = 3). Loading controls: tubulin and Ponceau red staining. **E** Left panels: Representative cell images of CHMP2B K6me1 (green) and alpha-tubulin (magenta) immunostaining in HeLa cells treated with DMSO or 10 uM of BAY-598. The white circle shows the midbody. Scale bar = 5 um. Right panel: quantification of the CHMP2B K6me1 signal at the midbody of HeLa cells treated with Ctrl or SMYD2 siRNAs as indicated (*N* = 3, *n* = 121 for DMSO and *n* = 125 for BAY-598 midbodies quantified by condition, mean ± SD, Two-sided Mann–Whitney test).

immunofluorescence analysis. The ICB signal was reduced in experiments using CHMP2B siRNA, supporting the antibody's specificity for CHMP2B (Supplementary Fig. 2C, D). The signal intensity significantly decreased with either SMYD2 silencing or pharmacological inhibition (Fig. 2C–E and Supplementary Fig. 2B), demonstrating its specificity for methylation. Moreover, while the signal at the ICB increased upon SMYD2 expression in wild-type (WT) CHMP2B_GFP expressing cells, this effect was not observed in CHMP2B K6A or CHMP2B K6R mutants expressing cells (Supplementary Fig. 2E). Thus, CHMP2B K6 methylation at the midbody is dependent on SMYD2 catalytic activity and could play a role during cytokinesis. Importantly, the K6 lysine residue is highly conserved in CHMP2 across evolution, including in yeast, which suggests its functional importance (Supplementary Fig. 2F). The CHMP2B K6 residue is positioned at the interface between a hydrophobic and hydrophilic region. The addition of a methyl group to this region could thus locally enhance the hydrophobicity potentially leading to a conformational change and CHMP2B activation (Supplementary Fig. 2G). Collectively, these findings demonstrate the presence of *in cellulo* methylation of endogenous CHMP2B at the intercellular bridge of cells undergoing cytokinesis and suggest an unexplored contribution of SMYD2 to the late cytokinetic steps.

## CHMP2B K6 is involved in the timely progression of CHMP2B to the abscission site
To evaluate the impact of SMYD2-mediated methylation on CHMP2B function during cytokinesis, we established stable HeLa cell lines

expressing RNAi-resistant (siR) forms of CHMP2B_GFP or an un-methylatable mutant, CHMP2B K6A_GFP (Fig. 3A). These cell lines exhibited comparable expression levels of CHMP2B to endogenous levels and displayed similar localization patterns at the midbody (Supplementary Fig. 3A, B). To assess the effect on abscission kinetics, we performed time-lapse phase contrast microscopy to monitor ICB scission upon CHMP2B silencing. Our time-lapse phase contrast microscopy analysis revealed a delay in abscission upon depletion of endogenous CHMP2B (Fig. 3B, C). Reintroduction of WT CHMP2B rescued the abscission timing, while reintroduction of the un-methylatable CHMP2B K6A mutant failed to rescue the cytokinetic

defect (Fig. 3B, C and Supplementary Movies 2–5). Indeed, the CHMP2B K6A cells exhibited slower resolution of their intercellular bridges compared to control cells, underscoring the crucial role of CHMP2B lysine 6 in regulating the timing of abscission during cell division (Fig. 3B, C). We observed an increased proportion of ICBs with CHMP2B K6A_GFP localized at ring structures at the midbody and a comparatively reduced proportion of ICBs with CHMP2B K6A_GFP localized at the midbody arms (compared to wild-type CHMP2B) (Supplementary Fig. 3C). We confirmed this observation in independent clones (Supplementary Fig. 3D, E). To investigate whether the delayed abscission observed in cells expressing CHMP2B K6A_GFP was

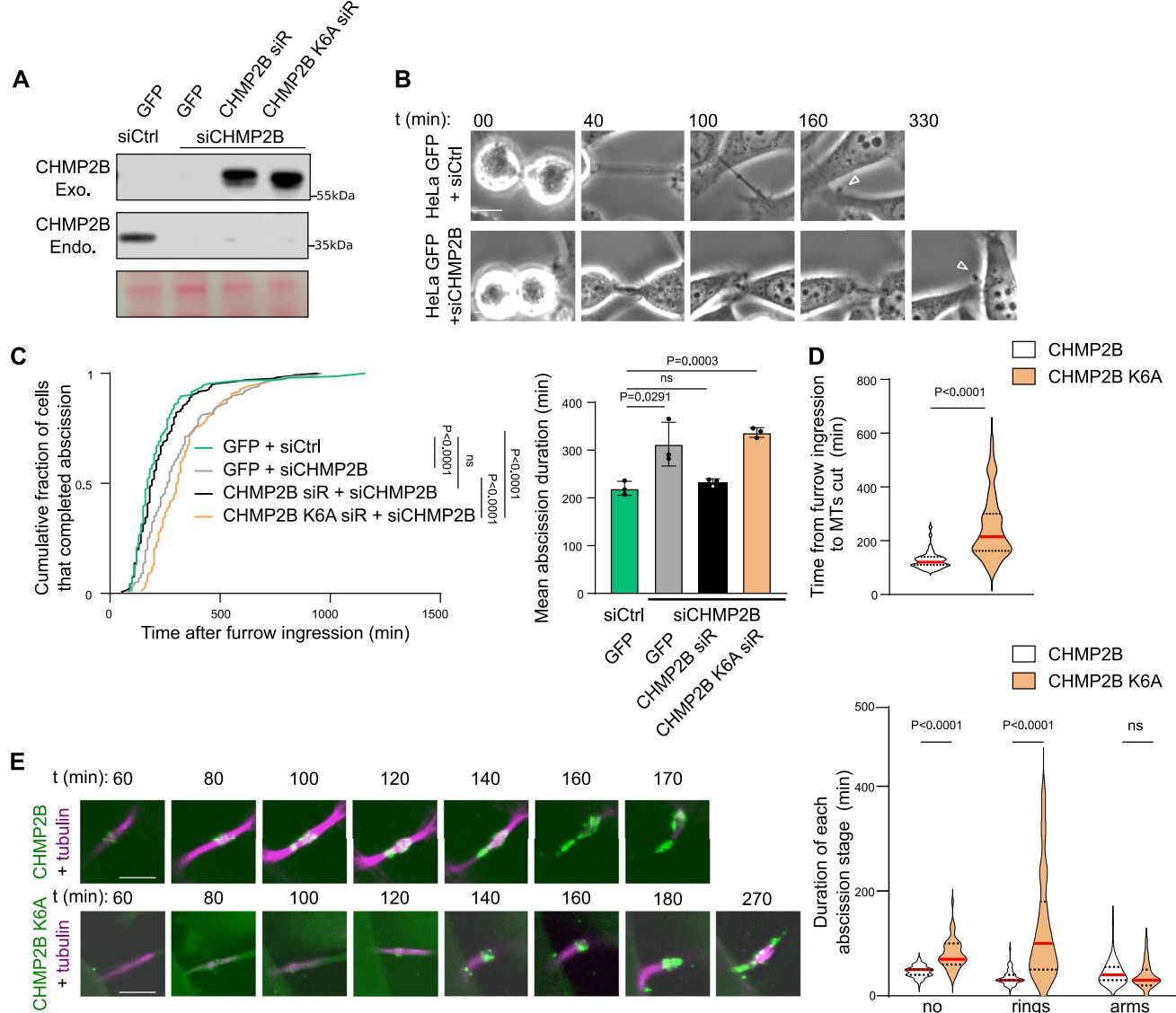

**Fig. 3 | CHMP2B K6 is involved in the timely progression of CHMP2B to the abscission site. A** Western blots of protein extracts from HeLa cell lines stably expressing GFP or siRNA resistant (siR) CHMP2B_GFP or CHMP2B K6A_GFP treated with control or CHMP2B siRNAs revealed with CHMP2B antibody (N = 3, CHMP2B exo. = CHMP2B_GFP protein, CHMP2B endo. = endogenous CHMP2B). Loading control: Ponceau red staining. **B** Selected frames from phase-contrast time-lapse videomicroscopy of HeLa cells stably expressing GFP treated with control or CHMP2B siRNAs entering in cytokinesis and completing their abscission. Time is expressed in minutes, and time 00 is set at the time point prior the intercellular bridge formation. Last time point corresponds to the ICB cut indicated by the empty arrow, considered here as the abscission (N = 3). Scale bar = 10 um. **C** Distribution of the abscission time measured by phase-contrast time-lapse microscopy in the indicated cells (N = 3, n > 109 cells, Two-sided Kolmogorov–Smirnov test, ns non-

significant (p > 0.05)) and the mean abscission duration (minutes, N = 3, mean ± SD, t-test, ns non-significant (p > 0.05)) are shown. **D** Quantification of the abscission time of HeLa cell lines stably expressing CHMP2B_GFP or CHMP2B K6A_GFP (from furrow onset to the microtubule cut), each dot represents one dividing cells. (N = 3, n > 50, median (red lines) and quartiles (dotty lines), Two-sided Mann–Whitney test). **E** Left panel: live cell imaging of HeLa cell lines stably expressing CHMP2B_GFP (top) or CHMP2B K6A_GFP (bottom) treated with siR-tubulin (magenta). Time is expressed in minutes, and time 00 is set at the time point prior the intercellular bridge formation. The last time point corresponds to the microtubule cut, considered here as the abscission. Scale bar = 5 um. Right panel: duration of each abscission stage measured from live cell imaging performed in CHMP2B_GFP or CHMP2B K6A_GFP expressing cells (N = 3, n > 50, median (red lines) and quartiles (dotty lines), Two-sided Mann–Whitney test, ns non-significant (p > 0.05)).

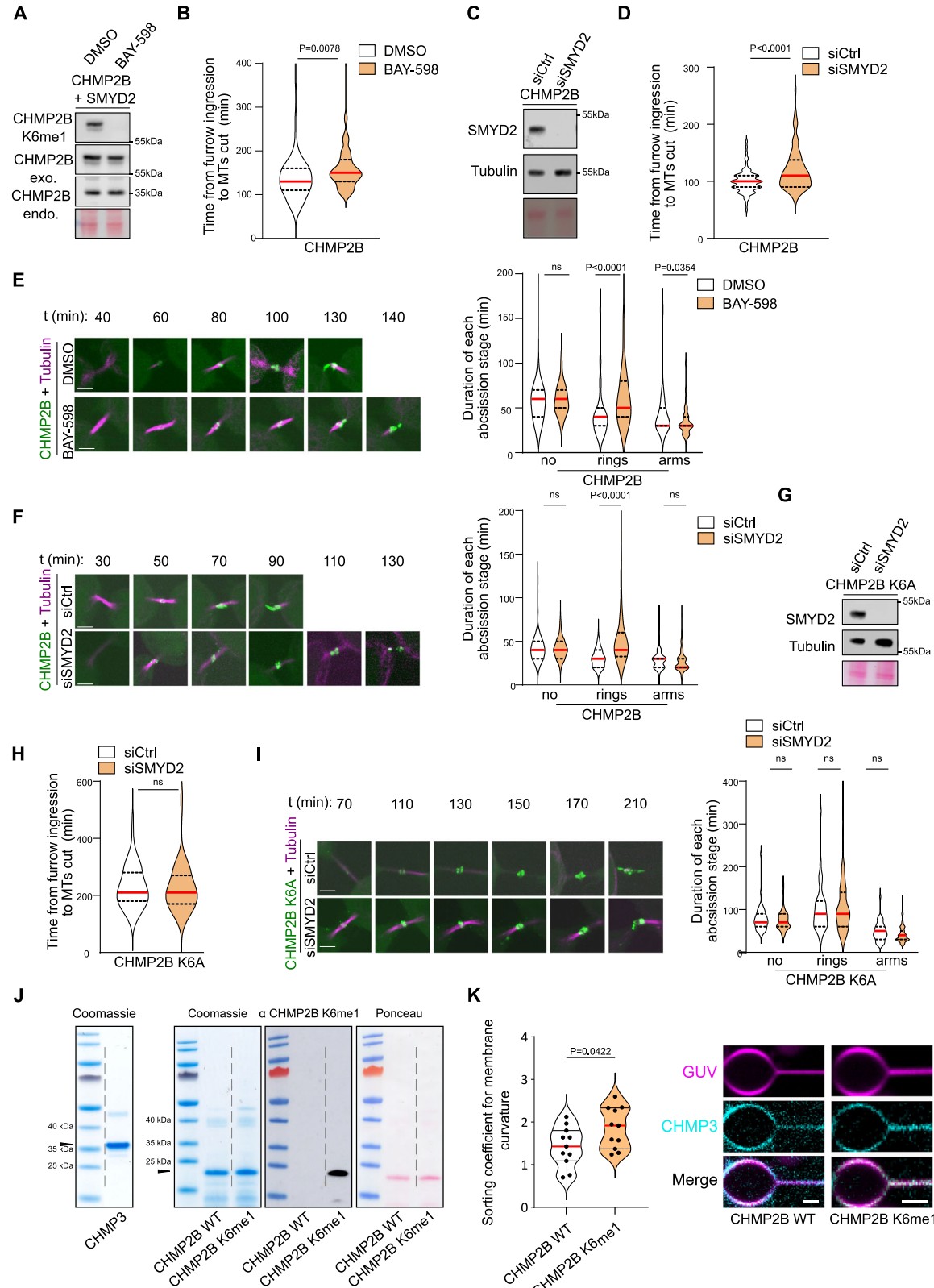

due to defective recruitment at the intercellular bridge, we employed time-lapse spinning-disk confocal microscopy. We categorized three stages of abscission based on CHMP2B localization: early abscission stage with no CHMP2B at the ICB, CHMP2B recruitment as midbody rings, and CHMP2B at midbody arms (illustrated in Supplementary Fig. 3A). Time-lapse microscopy confirmed the delayed abscission timing in cells expressing CHMP2B K6A_GFP compared to

CHMP2B_GFP, monitored by the microtubule cut (MT cut) (Fig. 3D and Supplementary Movies 6 and 7), which was consistent with results observed by phase-contrast microscopy (Fig. 3B, C). This delay was associated with slower recruitment of CHMP2B K6A_GFP at the midbody and reduced progression to the abscission site compared to CHMP2B_GFP (Fig. 3E and Supplementary Movies 6 and 7). To confirm these results, we generated stable HeLa cell lines expressing a CHMP2B

**Fig. 4 | SMYD2 regulates the localization of CHMP2B and the timing of abscission. A** Western blots of protein extracts from HeLa cell lines stably expressing CHMP2B_GFP and Cherry_SMYD2 treated for 24 h with DMSO or SMYD2 inhibitor (BAY-598 10 μM) were blotted for CHMP2B K6me1 or CHMP2B antibodies ($N = 3$). Loading control: Ponceau red. **B** Abscission time (from furrow onset to the microtubule cut) of HeLa cell line stably expressing CHMP2B_GFP and treated as indicated. ($N = 3$, $n > 65$ cells, median (red lines) and quartiles (dotty lines), Two-sided Mann–Whitney test). **C** Western blots of protein extracts from HeLa cell line stably expressing CHMP2B_GFP treated with control or SMYD2 siRNAs were blotted as indicated ($N = 3$). Loading control: Tubulin and Ponceau red. **D** Abscission time was determined as in (**B**). ($N = 3$, $n > 75$, median (red lines) and quartiles (dotty lines), Mann–Whitney test). **E** Left panels: live cell imaging of CHMP2B_GFP HeLa cell line treated as indicated. Time 00 is set prior the intercellular bridge formation. Last time point corresponds to the microtubule cut. Scale bar = 5 um. Right panel: Duration of each abscission stage ($N = 3$, $n > 65$ cells, median (red lines) and quartiles (dotty lines), Two-sided Mann–Whitney test, ns non-significant ($p > 0.05$)). **F** Left panels: live cell imaging of CHMP2B_GFP HeLa cell line treated as indicated. Time 00 is set prior the intercellular bridge formation. Last time point corresponds to the microtubule cut. Scale bar = 5 um. Right panel: Duration of each abscission stage ($N = 3$, $n > 75$ cells, median, Two-sided Mann–Whitney test, ns non-significant ($p > 0.05$)). **G** Western blots of protein extracts from CHMP2B K6A_GFP HeLa cell line treated with control or SMYD2 siRNAs were blotted as indicated ($N = 3$). Loading controls: Tubulin and Ponceau red staining. **H** Abscission time of CHMP2B K6A_GFP HeLa cell line treated with control or SMYD2 siRNAs was determined as in (**B**). ($N = 3$, $n > 41$ cells, median (red lines) and quartiles (dotty lines), Two-sided Mann–Whitney test, ns non-significant ($p > 0.05$). **I** Left panels: live cell imaging of CHMP2B K6A_GFP HeLa cell line treated as indicated. Time 00 is set prior the intercellular bridge formation. Last time point corresponds to the microtubule cut. Scale bar = 5 um. Right panel: Duration of each abscission stage ($N = 3$, $n > 41$ cells, median (red lines) and quartiles (dotty lines), Two-sided Mann–Whitney test, ns non-significant ($p > 0.05$)). **J** Recombinant CHMP3, CHMP2B-ΔC and CHMP2B-ΔC K6me1 proteins were purified and visualized using Coomassie (left) and Ponceau red staining (right). CHMP2B K6 methylation was assessed by western blot using CHMP2B K6me1 antibody ($N = 2$). **K** Left panel: Sorting coefficient plot showing preferential assembly of fluorescently labeled CHMP3 on positively curved membranes when in the presence of methylated CHMP2B. ($N = 2$, median (red lines) and quartiles (black lines), Two-sided $t$-test, ($p < 0.05$)). Right panel: Representative fluorescence micrograph used to calculate sorting coefficient values showing recruitment of fluorescently labeled CHMP3 on pulled lipid tubes from GUVs. Magenta is membrane signal, cyan is CHMP3 signal. Scale bar = 5 um.

K6R_GFP mutant, where lysine 6 is substituted with arginine, a mutation that conserves the residue's charge (Supplementary Fig. 3F). Using time-lapse spinning-disk confocal microscopy, we show that the CHMP2B K6R mutant also resulted in delayed abscission, slower recruitment, and reduced progression to the abscission site (Supplementary Fig. 3G, H). Taken together, these results demonstrate the significance of CHMP2B lysine 6 in both the recruitment of CHMP2B at the midbody and its transition to the abscission site during cytokinesis.

## SMYD2 regulates the localization of CHMP2B and the timing of abscission

We next investigated whether SMYD2-mediated methylation of CHMP2B altered ESCRT dynamics at the intercellular bridge. Silencing SMYD2 or the chemical inhibition of SMYD2 (BAY-598 drug) markedly decreased CHMP2B methylation (Figs. 4A and 2C–E) and resulted in a significant delay in the timing of abscission, observed by monitoring the microtubule cut (Fig. 4B–D). In addition, SMYD2 loss-of-function (genetically or pharmacologically) increased the retention of CHMP2B_GFP at ring structures at the midbody without impacting the CHMP2B_GFP initial recruitment at the ICB (Fig. 4E, F and Supplementary Movies 8–11). These results partially phenocopied the observations with the methyl-resistant CHMP2B K6A (Fig. 3E) and CHMP2B K6R mutants (Supplementary Fig. 3H). The difference in amplitude observed between the use of CHMP2B K6 mutants and SYMD2 depletion/inhibition suggests that mutating the K6 residue has a stronger physicochemical impact than inhibiting its methylation. We monitored the localization of endogenous CHMP2B at the ICB at different stages of abscission. SMYD2 silencing also increased the retention of endogenous CHMP2B at the ring structures of the midbody, compared to the control condition (Supplementary Fig. 4A, B). We observed similar results in U2OS osteosarcoma cells indicating a general requirement of SMYD2 for CHMP2B proper localization at the ICB in different cell types undergoing cytokinesis (Supplementary Fig. 4C, D). We then investigated whether these findings were dependent on the K6 residue methylated by SMYD2. Interestingly, while SMYD2 depletion increased the retention of CHMP2B_GFP at ring structures and caused a delay in abscission, it did not delay abscission or affect the localization of the un-methylatable CHMP2B K6A_GFP mutant (Fig. 4G–I and Supplementary Movies 12 and 13). This is consistent with SMYD2 regulating the localization of CHMP2B at the ICB and the timing of abscission through methylation of the K6 residue. Altogether, these results suggest that methylation of CHMP2B regulates the timing of abscission by controlling CHMP2B polymerization and transition toward the site of abscission.

To explore the potential mechanism responsible for the *in cellulo* effect observed with the methylated CHMP2B, we turned to in vitro reconstitution systems using purified ESCRT-III proteins and model membranes. CHMP2B lacking the C-terminal region (CHMP2BΔC) was previously shown to increase the efficiency of CHMP3 binding to membranes[36,37]. We analyzed the effect of CHMP2BΔC methylation on the recruitment of CHMP3. We compared unmethylated CHMP2B-ΔC with CHMP2B-ΔC that was methylated in vitro by SMYD2 (70% methylation of K6 residue) (Fig. 4J and Supplementary Fig. 4E). To monitor CHMP2B-ΔC/CHMP3 recruitment on curved membranes, we chemically labeled CHMP3 to circumvent the aggregation issues encountered with CHMP2B-ΔC during the labeling process (Fig. 4J). To experimentally obtain curved membranes, we pulled membrane nanotubes from giant unilamellar vesicles (GUVs) dopped with Atto-647 fluorochrome. This allowed us to study the effect of CHMP2B methylation on CHMP3 recruitment and polymerization on curved membranes, as previously shown for unmethylated CHMP2A-ΔC/ CHMP3 complex[38]. We measured the fluorescence intensity of CHMP3 normalized to the membrane fluorescence intensity on both the tube (curved) and the GUV (flat), and calculated the sorting coefficient as the ratio between them. CHMP2B-ΔC/CHMP3 complex showed stronger preference for positively curved membranes as compared with the CHMP3 protein alone that was not able to bind by itself to the curved membrane. Interestingly, we found that a higher sorting coefficient value was obtained with the methylated CHMP2B-ΔC/CHMP3 complex (Fig. 4K), showing that methylation may promote the formation of curvature-sensing filaments, either because they a have a smaller preferred curvature, or because they are more rigid. These results suggest that CHMP2B-ΔC K6 methylation facilitates the formation of ESCRT-III filaments on curved membrane.

## SMYD2 accelerates abscission timing when cytokinesis is challenged

SMYD2 is overexpressed in several types of cancers[25]. However, additional research is required to gain a comprehensive understanding of SMYD2's role in cell division and the development of tumors. To investigate the impact of SMYD2-dependent CHMP2B hyper-methylation, we established stable HeLa cell lines expressing CHMP2B_GFP and co-expressing Cherry_SMYD2. The ectopic expression of Cherry_SMYD2 increased the methylation level of CHMP2B in total cell extracts (Supplementary Fig. 5A) and at the intercellular bridge (Supplementary Fig. 5B) without altering the localization of CHMP2B at the ICB (Supplementary Fig. 5C, D and Supplementary Movies 14 and 15). Next, we monitored the abscission timing in HeLa cell lines stably

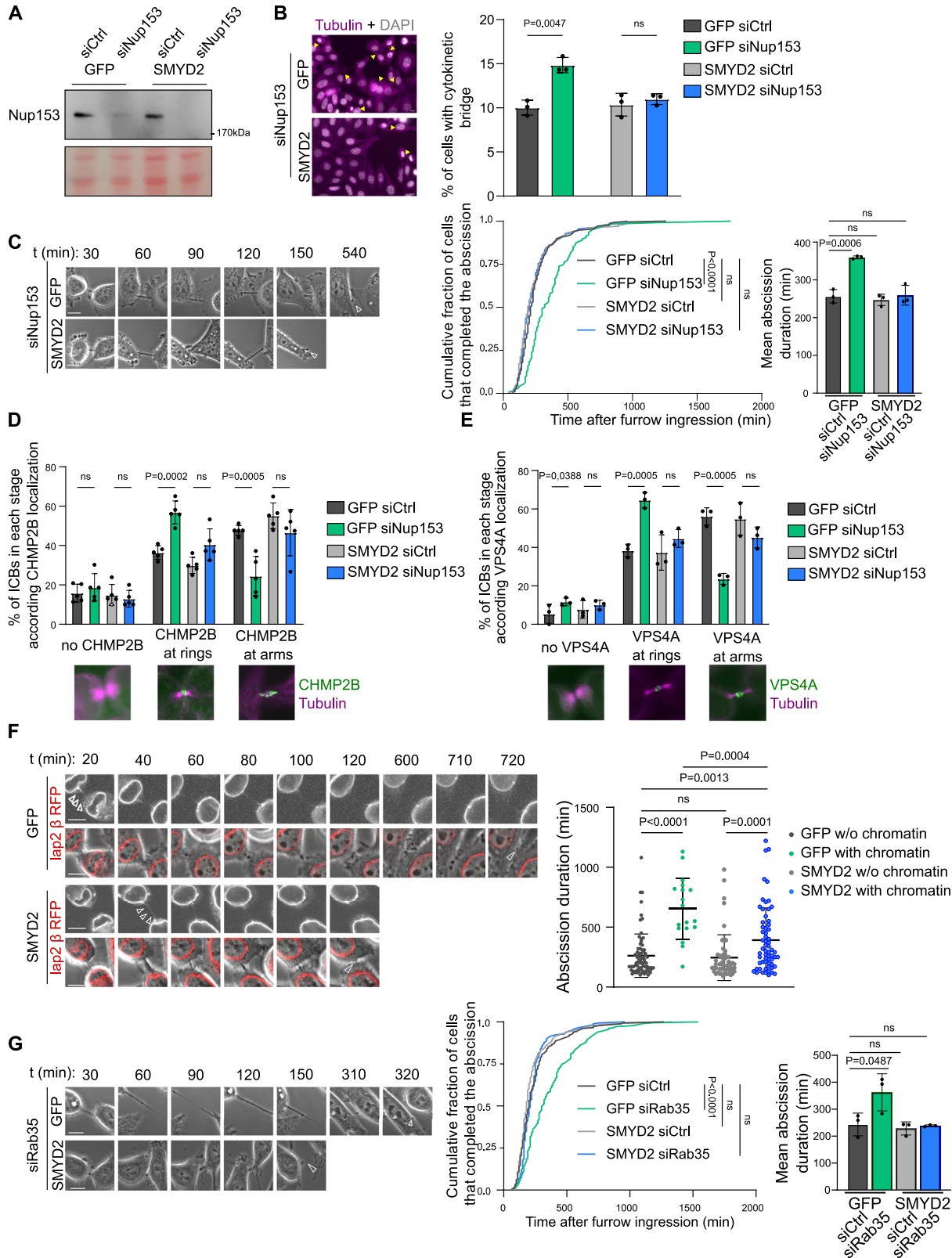

expressing SMYD2_GFP that also efficiently increased CHMP2B K6 methylation levels at the ICB (Supplementary Fig. 5E, F). Although we observed no difference in the overall abscission timing (Supplementary Fig. 5G), we found that SMYD2 accelerated abscission in cells with slower abscission kinetics (Supplementary Fig. 5H). This finding suggested that when the abscission process is delayed, SMYD2 can accelerate the abscission process. To investigate whether CHMP2B

methylation promotes abscission in cells undergoing cytokinetic challenges that lead to abscission delay, we activated the abscission checkpoint by partially depleting Nucleoporin 153 (Nup153) in HeLa cells expressing SMYD2_GFP (Fig. 5A). Depletion of Nup153 increased the number of cells connected by an ICB in control GFP cells, as previously reported[39] (Fig. 5B). Remarkably, this effect was not observed in HeLa cells expressing ectopic SMYD2_GFP, suggesting that SMYD2

**Fig. 5 | SMYD2 accelerates abscission timing when cytokinesis is challenged.**
**A** Western blots of protein extracts from HeLa cell lines stably expressing GFP or
SMYD2_GFP treated with control or Nup153 siRNAs ($N = 3$). Loading control: Ponceau red. **B** Left panels: representative cell images. Yellow arrows show cytokinetic cells connected by an ICB. Scale bar = 10 um. Right panel: quantification of cytokinetic cells ($N = 3$, $n > 300$ cells, mean ± SD, Two-sided multiple unpaired $t$-test, ns non-significant ($p > 0.05$)). **C** Left panels: frames of the indicated HeLa cell lines. Time 00 is set prior the ICB formation. Last time point corresponds to the ICB cut, indicated by the arrow. Scale bar = 10 um. Right panel: distribution of the abscission time measured by phase-contrast time-lapse microscopy ($N = 3$, $n > 126$ cells, Two-sided Kolmogorov–Smirnov test, ns non-significant ($p > 0.05$)) and mean abscission duration ($N = 3$, mean ± SD, $t$-test, ns non-significant ($p > 0.05$)) are shown.
**D** Quantification of ICBs with either no CHMP2B, CHMP2B in rings at the midbody, and CHMP2B both at the midbody and at the abscission site ($N = 5$, $n > 40$ ICBs counted/$N$, mean ± SD, Two-sided multiple unpaired $t$-test, ns non-significant ($p > 0.05$)). **E** Quantification of ICBs with either no VPS4A, VPS4A at midbody rings,

and VPS4A both at the midbody and at the abscission site ($N = 3$, $n > 50$ ICBs counted/$N$, mean ± SD, Two-sided multiple unpaired $t$-test, ns non-significant ($p > 0.05$)). **F** Left panels: selected phase contrast time-lapse and fluorescent videomicroscopy frames of HeLa cell lines with Lap2_β_RFP that stains for chromatin bridges (indicated by 3 arrows). Last time point corresponds to the ICB cut. Scale bar = 10 um. Right panel: distribution of the abscission time measured by phase-contrast time-lapse microscopy plotted as individual points. ($n = 74$, $n = 18$, for GFP cells with or without chromatin bridge and $n = 53$; $n = 62$ for GFP-SMYD2 cells with or without chromatin bridge, $N = 3$ experiments, mean ± SD, Two-sided Mann–Whitney test, ns non-significant ($p > 0.05$)). **G** Left panels: selected frames of HeLa cell lines treated with Rab35 siRNA. Time 00 is set prior the ICB formation. Last time point corresponds to the ICB cut. Scale bar = 10 um. Right panel: distribution of the abscission time measured by phase-contrast time-lapse microscopy ($N = 3$, $n > 166$ cells, Two-sided Kolmogorov–Smirnov test, ns non-significant ($p > 0.05$)) and mean abscission duration ($N = 3$, mean ± SD, $t$-test, ns non-significant ($p > 0.05$)).

overexpression rescued the abscission delay induced by the abscission checkpoint (Fig. 5B). This result was further confirmed by time-lapse, phase-contrast microscopy analysis (Fig. 5C and Supplementary Movies 16 and 17). While CHMP2B and VPS4A were retained at the midbody center in cells depleted for Nup153, ectopic SMYD2 expression promoted CHMP2B and VPS4A progression toward the abscission site (Fig. 5D, E and Supplementary Fig. 6A, B). Notably, the rescue phenotype was dependent on SMYD2 catalytic activity, as expression of the SMYD2 Y240A mutant failed to rescue (1) the abscission delay (Supplementary Fig. 6C) and (2) the retention of CHMP2B and VPS4 at the midbody center (Supplementary Fig. 6D, E). These findings suggested that SMYD2 induced methylation of ESCRT-III bypasses the abscission checkpoint delay by promoting CHMP2B polymerization at the abscission site. We further investigated whether SMYD2 overexpression could bypass the abscission checkpoint in the context of chromatin bridge formation. We generated HeLa cell lines that stably express the Lap2-β protein tagged with RFP to visualize chromatin bridges. We observed a significant delay in abscission in cells with chromatin bridges by time-lapse microscopy analysis, which was substantially reduced by ectopic expression of SMYD2 (Fig. 5F and Supplementary Movies 18 and 19). Finally, we investigated whether the promotion of abscission by SMYD2 was restricted to the abscission checkpoint context, or occurred when the abscission delay was induced by an independent mechanism. We induced abscission delay by depleting the GTPase Rab35 protein as previously described[40–43]. Once again, we found that ectopic expression of SMYD2 promoted the abscission process and rescued the abscission delay (Fig. 5G, Supplementary Fig. 6F and Supplementary Movies 20 and 21). Taken together, these results demonstrate that SMYD2 methylation of ESCRT-III rescued the abscission delay in three different contexts: (1) upon activation of the checkpoint due to nuclear pore defects (siNup153), (2) in response to chromatin bridges trapped in the ICB, and (3) upon Rab35 depletion.

### CHMP2B K6 and its methylation contribute to HIV particles release

ESCRT-III proteins play a crucial role in diverse membrane remodeling processes, including HIV-1 budding, particularly involving the CHMP2 proteins[44]. To extend our understanding on the function of the CHMP2B K6 methylation in broader processes, such as HIV budding, we monitored the impact of the CHMP2B K6A and CHMP2B K6R mutants in a single round of HIV-1 infection. HeLa cells expressing either the WT or the un-methylatable (K6A/R) CHMP2B_GFP were transfected with HIV-1 WT provirus. After allowing viral proteins to accumulate for 48 h, we measured the viral titer and levels of cell-associated and released CAp24. Cells expressing the un-methylatable CHMP2B K6A/R exhibited a decrease in CAp24 release into the supernatant compared to cells expressing GFP alone (Fig. 6A), which

led to a reduction in the release of HIV-1 WT (Fig. 6B). Moreover, CHMP2B K6A and CHMP2B K6R inhibited the production of infectious virions (Fig. 6C). A previous study showed that simultaneous silencing of both CHMP2B and CHMP2A is necessary to interrupt virus budding[44], highlighting the functional redundancy between these two members of the CHMP2 family. The substantial decrease in virus release that we observed with the CHMP2B K6A/R mutants could be the result of a dominant-negative effect via co-polymerization with endogenous ESCRT-III members, including CHMP2A. To explore the impact of K6 methylation on HIV-1 release, we investigated the silencing of SMYD2. The depletion of SMYD2 significantly decreased the release of HIV-1 WT (Fig. 6D, E) and inhibited the production of infectious virions (Fig. 6F). These results collectively suggest that the function of CHMP2B K6 and its methylation are not limited to ESCRT-III functions in abscission, but also play an important role in ESCRT-III-dependent HIV budding and membrane remodeling processes in general.

## Discussion

Here, we report the discovery of a post-translational modification critical for controlling the kinetics of membrane scission. We link a single methylation event of a component of the membrane remodeling ESCRT-III machinery with the timing of cytokinetic abscission and viral budding. Our study demonstrates that the CHMP2B ESCRT-III component undergoes methylation by the SMYD2 lysine methyltransferase both in vitro and in vivo, specifically on the lysine 6 (K6) residue, which is highly conserved across different organisms (Supplementary Fig. 2F). A specific antibody recognizing CHMP2B K6 me1 provided direct experimental evidence for the SMYD2-dependent methylation of CHMP2B in vivo at the intercellular bridge of cytokinetic cells. Strikingly, expression of un-methylatable CHMP2B mutants (CHMP2B K6A, CHMP2B K6R) perturbed the localization of ESCRT-III at the ICB with delayed recruitment and distribution of CHMP2B to the cleavage site, ultimately leading to impeded abscission. The dominant-negative effect we observed with the CHMP2B K6A/R mutants could be due to its co-polymerization with endogenous ESCRT-III members (including CHMP2A, CHMP2B, CHMP3, or CHMP4B) which could retain the ESCRT-III machinery along the ICB. Gain- and loss-of-function of SMYD2 led to substantial changes in CHMP2B methylation at the ICB. The hypomethylation of CHMP2B (from genetic silencing or pharmacological inhibition of SMYD2) caused a delay in the polymerization of CHMP2B at the abscission site, resulting in abscission hindrance. Conversely, the hypermethylation of CHMP2B upon SMYD2 increased expression resulted in an accelerated abscission. These findings suggest that methylation functions as a molecular timer mechanism for ESCRT-III abscission.

Our results raise the mechanistic question of how lysine methylation might impact the activity of ESCRT-III proteins. We speculate

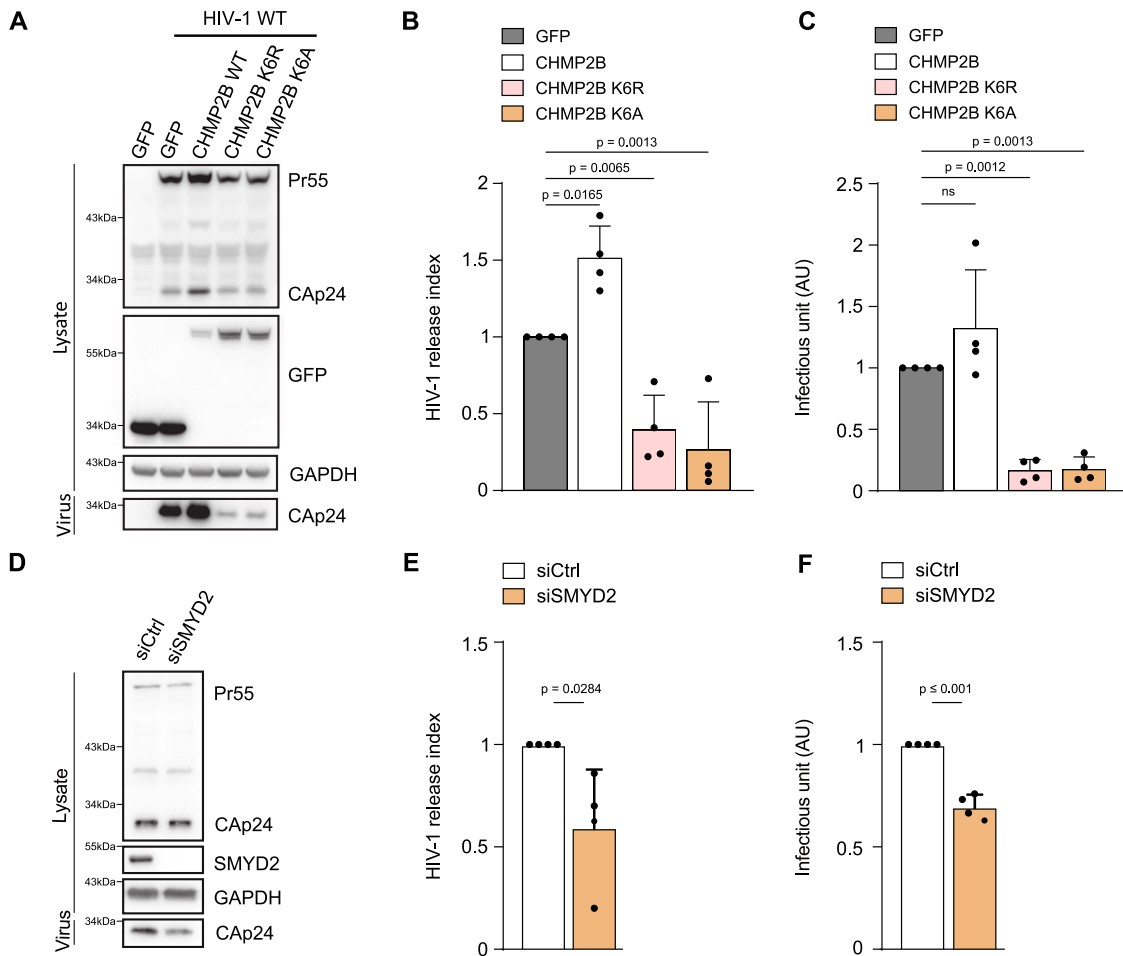

**Fig. 6 | CHMP2B K6 and its methylation contribute to HIV particles release.**
HeLa cells expressing GFP, CHMP2B_WT-GFP, CHMP2B K6R-GFP and CHMP2B K6A-GFP were transfected with pNL4-3 HIV-1 pro-viral DNA. Forty-eight hours later, cells were lysed and virus contained in supernatants were pelleted by ultracentrifugation. Viral production was evaluated by western blot and by ELISA quantification of HIV-1 CAp24. **A** Western blot analysis of HIV-1 Gag and CAp24 products, GFP and GAPDH in transfected cells and pelleted viruses. **B** HIV-1 release index was calculated as the ratio between released CAp24 and cell-associated CAp24. Results were normalized to control cells (set as 100%). **C** The virus titer was scored by infection of HeLa P4R6 indicator cells, followed by β-galactosidase activity quantification in the cells. (Statistical analysis using two-tailed unpaired Student's $t$ test, ns non-significant ($p > 0.05$), $N = 4$, mean ± SD. All western blots are representative of at least

three independent experiments). **D** HeLa cells transfected with indicated siRNA were transfected with pNL4-3 HIV-1 pro-viral DNA. Forty-eight hours later, cells were lysed, and virus contained in supernatants were pelleted by ultracentrifugation. Viral production was evaluated by western blot and by ELISA quantification of HIV-1 CAp24. Western blot analysis of HIV-1 Gag and CAp24 products, SMYD2 and GAPDH in transfected cells and purified viruses. **E** HIV-1 release index was calculated as the ratio between released CAp24 and cell-associated CAp24. Results were normalized to control cells (set as 100%). **F** The virus titer was scored by infection of HeLa P4R6 indicator cells, followed by β-galactosidase activity (Statistical analysis using two-tailed unpaired Student's $t$ test, $N = 4$, mean ± SD. All western blots are representative of at least three independent experiments).

---

that methylation could alter various molecular aspects of ESCRT-III function, such as membrane insertion, hetero-polymerization, and/or inter-filament interactions. Methylation occurs in the N-terminus of CHMP2B which has been linked to effective functioning of CHMP2B in various cellular processes[45]. ESCRT-III proteins are maintained in an autoinhibited state within the cytosol, through a folded structure between the N-terminal and C-terminal domains. Upon interaction with the membrane, this inhibitory state is released, allowing the formation of polymers in the form of spirals or helical tubular structures[46–49]. K6 methylation might modulate the dynamic properties and topology of ESCRT-III by altering the hydrophobic and steric properties of CHMP2B. This could lead to changes in intramolecular interactions and/or ESCRT-subunit interactions, thereby facilitating the activation of ESCRT-III at the intercellular bridge. Alternatively, K6 methylation might directly affect the membrane anchoring of CHMP2B. In mammals, the CHMP2B N-terminus is important for association with the plasma membrane, allowing it to polymerize and induce membrane tubulation[38,45]. Structural studies revealed that the

N-terminus of CHMP2A, a closely-related protein to CHMP2B, is positioned for membrane insertion[50]. The CHMP2B K6 methylation site falls within a domain described as essential for CHMP2B's ability to bind and deform membranes[45,46]. Specifically, the methylated CHMP2B K6 is located in a short amphipathic N-terminal unstructured sequence at the junction between two regions that exhibit differential hydrophobic states (Supplementary Fig. 2G). Methylation could increase the hydrophobicity of the CHMP2B N-terminus, thereby increasing binding to the hydrophobic plasma membrane[45] or strengthening the interaction with the membrane and promoting its polymerization. Reconstitution experiments in vitro highlighted the importance of a continuous and ordered exchange of ESCRT-III subunits to drive membrane scission[10,12] with CHMP4 and CHMP2 yeast orthologs (SNF7 and VPS2, respectively) playing a critical role. Using in vitro reconstitution systems on model membranes, we found that CHMP2B methylation increased CHMP3 binding to curved membranes and could thus facilitate the formation of curvature-sensing ESCRT-III filaments. This observation provides insight into the molecular

mechanism underlying the *in cellulo* phenotypes observed; methylation impacts membrane interactions and recruitment which promotes the relocalization of CHMP2B to the abscission site and thereby favors the abscission process.

We found that SMYD2-dependent CHMP2B methylation accelerates abscission timing in several cellular scenarios. When cells experience an abscission checkpoint (e.g., triggered by nuclear pore defects or chromatin bridge formation) or a slowdown in the abscission process independently of the abscission checkpoint (e.g., linked to factors like Rab35), CHMP2B and other ESCRT-III members are halted and unable to polymerize toward the site of abscission. In this context, the overexpression of SMYD2 effectively bypassed the delay in abscission, allowing ESCRT-III to transition toward the cleavage site where membrane scission occurs. Remarkably, ectopic expression of SMYD2 resulted in the hypermethylation of CHMP2B, potentially enhancing membrane scission by facilitating the formation of spiral-like filaments. Our functional live-cell microscopy imaging suggested that the methylation of CHMP2B is a key PTM that accelerates abscission by promoting CHMP2B relocation at the abscission site. This discovery significantly contributes to our understanding of how post-translational modifications can regulate the functions of ESCRT-III, ultimately influencing the timing and efficiency of the scission process.

Our study raises intriguing functional questions about the overexpression of SMYD2 in various cancer types. In normal cells, ectopic expression of SMYD2 had a modest effect, but this became more pronounced when cells faced cytokinetic challenges which are common in cancer cells. SMYD2 is upregulated in many types of cancer and is associated with a marked decrease in overall survival and accelerated disease progression[25]. The widespread pro-oncogenic role of SMYD2 suggests that this methyltransferase may regulate a common biological process that contributes to tumor development across a diverse range of cancers. Normal cells tightly regulate the cytokinetic process (through the abscission checkpoint) to ensure the faithful inheritance of genetic material during cell division. Compelling evidence is emerging which highlights the importance of phosphorylation in this process. For example, upon encountering cytokinetic challenges (such as chromatin entrapment at the ICB), Aurora B kinase phosphorylates the CHMP4C subunit of ESCRT-III[17,51]. This phosphorylation, in conjunction with ANCHR, leads to the retention of the VPS4 ATPase at the midbody, resulting in delayed completion of abscission[52]. Phosphorylated CHMP4C can phase-separate with ESCRT-associated proteins, further contributing to the delay in abscission[53]. Additionally, the ULK3 kinase phosphorylates and inhibits IST1, an ESCRT-III component, thereby sustaining the abscission checkpoint[16]. While phosphorylation appears to maintain the abscission checkpoint, the specific PTMs that enable checkpoint exit remain unidentified. Here, we propose that methylation could trigger the premature exit from the abscission checkpoint in cancer cells with elevated SMYD2 expression. We observed that ectopic SMYD2 expression bypassed the abscission checkpoint and promoted ESCRT-III polymerization toward the abscission site leading to a premature cleavage of the ICB. We speculate that this premature abscission, coupled with insufficient time for cellular repair processes, could lead to aneuploidy and polyploidy, thereby contributing to cancer progression[54,55]. In this regard, recent studies have implicated ESCRT-III, specifically the CHMP4C subunit and VPS4, in cancer development[55–57]. Future studies will evaluate the CHMP2B methylation status in normal and cancer tissues to demonstrate the potential pro-oncogenic impact of ESCRT-III methylation signaling mediated by SMYD2.

This study unveiled the molecular and cellular significance of lysine methylation in cell division, specifically in regulating the precise spatiotemporal progression of ESCRT-III. Our observation that CHMP2B methylation also regulates HIV budding raises the compelling possibility that methylation might serve as a general mechanism for modulating ESCRT-III dynamics, extending its relevance to various ESCRT-dependent cellular processes, including plasma membrane and organelle repair, autophagosome closure, and nuclear envelope reformation.

# Methods

## Cell cultures

HeLa cells CCL-2 (American Type Culture Collection ATCC CCL-2, human cervical carcinoma, female) and U2OS cells (ATCC HTB-96, human osteosarcoma, female) were grown in Dulbecco's Modified Eagle Medium (DMEM; Life Technologies) supplemented with 10% fetal bovine serum and 1% penicillin–streptomycin in 5% $CO_2$ at 37 °C. HeLa cell lines stably expressing GFP, CHMP2B_GFP, CHMP2B K6A_GFP, SMYD2_GFP, SMYD2 Y240A_GFP, CHMP2B_GFP + Cherry and CHMP2B_GFP + Cherry_SMYD2 were generated by lentiviral transduction of HeLa CCL-2 with lentiviruses and selected by FACS sorting to obtain a bulk population of cells expressing the different fusion proteins at comparable level and near to the endogenous protein expression level (lentivectors are reported Supplementary Table 1). HeLa CHMP2B_GFP and CHMP2B K6A_GFP were then subject to clonal selection on 3 criteria: CHMP2B_GFP expression levels comparable to endogenous expression, similar expression level between CHMP2B_GFP and CHMP2B K6A_GFP, and CHMP2B localization at the midbody that recapitulates the endogenous one. CHMP2B_GFP and CHMP2B K6A_GFP are siRNAs resistant cell lines that have been obtained by mutating 6 nucleotides of the siRNA-targeting sequence using Phusion Site-Directed Mutagenesis Kit (Thermo Scientific). To generate Lap2_ β_RFP cell lines, GFP or SMYD2_GFP HeLa cells were transfected with pmRFP_ β_IRES_puro2b plasmid[58] (plasmid created by Dr Daniel Gerlich lab, IMBA, Vienna), followed by 1 µg/ml puromycin selection and cytometry sorting.

## Transfections and siRNAs

Plasmids were transfected in HeLa cells for 24 h using Lipofectamine 2000 Transfection Reagent (Thermo Scientific) or with Equilibrate X-tremeGENE 9 reagent (ref. 6365787001 Roche). For silencing experiments, siRNAs oligonucleotides were complexed with Lipofectamine RNAiMAX transfection reagent (Invitrogen) in Opti-MEM Reduced Serum Medium and HeLa or U2OS cells were plated on the mix in penicillin and streptomycin free DMEM, at a final siRNA concentration of 20 nM (Ctrl, CHMP2B, SMYD2 and Rab35 siRNAs) and 10 nM (Nup153 siRNAs). After 16h00, transfection media was changed and cells were maintained in DMEM with 10% fetal bovine serum and 1% penicillin–streptomycin for 48 h (for CHMP2B, SMYD2 and Rab35 depletion) or 72 h (for Nup153 depletion). Plasmids and siRNA are reported in Supplementary Tables 2 and 3.

## Plasmid constructs

Human SMYD2 and CHMP2B were subcloned into Gateway pDonor207 plasmid and GFP, Cherry or Flag transient expression vectors were generated by LR recombination (Thermo Fisher) into pDest-eGFP-C1, pDest-mCherry-C1 or pCiNeo-3xFLAG[59] destination vectors. Lentiviral vectors containing GFP, CHMP2B_GFP, CHMP2B K6A_GFP, CHMP2B K6R_GFP, SMYD2_GFP, SMYD2 Y240A_GFP, CHMP2B_GFP, Cherry or Cherry_SMYD2 were generated either by classical enzymes digestion followed by ligation, or by using in-fusion HD Cloning Kit (Takara bio). CHMP2B K6A, CHMP2B K6R and SMYD2 Y240A point mutations have been generated using Agilent QuikChange XL site-directed mutagenesis kit or Phusion Site-Directed Mutagenesis Kit. Complementary DNAs (cDNA) coding for the full length of human SMYD2, CHMP2B and CHMP3 proteins were obtained from HeLa total cDNA and subcloned into pSUMO (LifeSensor), pGEX-6P-1 (Addgene) and pet28 MBP TEV (Addgene) vectors respectively in order to produce 6xHis-tagged SUMO-SMYD2, GST-tagged CHMP2B, and 6xHis-tagged MBP-CHMP3 proteins. A shorter version of

CHMP2B (1-154) was also cloned in the pet28 MBP TEV vector in order to produce 6xHis-tagged MBP- CHMP2B 1-154 protein. BL21 HI-Control™ (DE3) *E.Coli* (for pSUMO and pGEX-6P-1 plasmids) and BL21 Rosetta 2™ (DE3) *E.Coli* (for pet28 MBP TEV plasmid) were then transformed with these plasmids in order to express and purify recombinant proteins. Primers used for mutagenesis are reported in Supplementary Table 4, and mutations was verified through DNA sequence analysis (Eurofins).

## Western blots

Total proteins were extracted in 1X Laemmli lysis buffer (prepared from Laemmli 2X: 4% SDS, 20% glycerol, 120 mM Tris HCl 1 M pH 6.8, 0,02% bromophenol blue, 10% β mercaptoethanol) and heated 10 min at 95° before loading. Protein extracts were loaded on NuPAGE Bis-Tris 4−12% Gels (Invitrogen) for migration, and transferred to nitrocellulose membranes. Protein transfer was assessed by Ponceau red staining. Membranes were blocked in PBS containing 0.1% Tween-20 and 5% milk for 1 h at room temperature. Incubations with primary antibodies were carried out at 4 °C overnight using the manufacturer recommended dilutions. After 3 washes in PBS-tween 0.1%, membranes were incubated with secondary antibodies coupled with peroxidase (Jackson ImmunoResearch) at room temperature for 1 h. Proteins were detected by chemiluminescence using SuperSignal (Thermo Fisher Scientific) with the LI-COR Odyssey FC Imaging System and Image Studio Lite Software.

For the analysis of HIV-1 production, cells were lysed using a cold DOC buffer (10 mM Tris, pH 8, 150 mM NaCl, 1 mM EDTA, 1% Triton X-100 and 0.1% DOC 10%) with complete protease inhibitor cocktail (Roche). The lysates were then centrifuged at 13,000 rpm for 15 min and protein concentrations were determined using a Bradford protein assay (Bio-Rad). The proteins were separated using SDS-PAGE, and were transferred to 0.45 μm PVDF membranes from Millipore, followed by blocking in milk buffer (Tris-buffered saline [TBS] [0.5 M Tris pH 8.4, 9% {wt/vol} NaCl], 5% [wt/vol] nonfat dry milk, 0.05% [vol/vol] Tween 20). Membranes were incubated overnight at 4 °C with the primary antibodies in milk buffer and incubated with appropriate HRP-conjugated secondary antibodies. After washing, protein bands were detected by using Amersham ECL Select Western blotting detection reagent (GE Healthcare). Antibodies used in this paper are reported in Supplementary Table 5.

## Protein-protein interaction studies

HeLa cells were lysed in lysis buffer (20 mM Tris HCl, pH 8, 150 mM NaCl, 0.4% to 1% NP40, 2 mM EDTA, protease inhibitors) 24 h post transfection, sonicated for 7 min and 30 s (10 s on, 50 s off), and protein lysates were collected after 20 min centrifugation at 13,000 rpm. Protein lysates were incubated O/N with GFP antibody Dynabeads magnetic beads (Thermo Fisher Scientific). After washing three times, the beads were heated at 95 °C in Lammeli 1 X buffer, and immunoprecipitated products were then analyzed by western blot.

## Immunofluorescence and image acquisition

HeLa or U2OS cells were grown on coverslips and then fixed with cold methanol for 5 min on ice. Cells were rinsed twice in PBS and then blocked for 30 min with PBS Tween 0.2%, 1% BSA and 1% SVF to prevent non-specific staining. After three washes in PBS Tween 0.2%, cells were incubated with primary antibodies in PBS Tween 0.2% at room temperature for 40 min−1 h. After three washes in PBS 0.2% Tween, cells were incubated for 30 min with secondary antibodies. After three washes in PBS Tween 0.2% and one last wash in water, coverslips were mounted with VECTASHIELD PLUS Antifade Mounting Medium with DAPI (EuroBio Scientific). Images were acquired with a fluorescent microscope Leica Inverted 6000B using a CCD camera (Photometrics) and Metamorph software. Images were analyzed with ImageJ software.

## Time-lapse microscopy

For time-lapse phase-contrast imaging, HeLa cells were plated on μ-Dish 35 mm Quad (ibidi), and maintained in a chamber at 37° with 5% CO₂ (Life Imaging Service). Time-lapse sequences were recorded every 10 min for 48 h using sCMOS Orca-Flash4 V2+ camera (Hamamatsu) on an inverted Leica DMI8 with HCX Plan APO 40x NA1,3 OIL PH3 CS objective and Metamorph software. For time-lapse fluorescent microscopy, images were acquired on cells treated with SiR-tubulin (spirochrome), using sCMOS Orca-Flash4 V2+ camera (Hamamatsu) on an inverted Leica DMI8 equipped with a CSU-W1 spinning disk confocal scanning unit (Yokogawa−Andor) or using EMCCD Camera (Evolve 512 Delta, Photometrics) inverted Eclipse TiE Nikon microscope equipped with a CSU-X1 spinning disk confocal scanning unit (Yokogawa). Images were acquired every 10 min overnight using HC PL APO 63x NA1.4 OIL CS2 or with a ×60 1.4 NA PL-APO VC objective and Metamorph software. Both for phase-contrast and fluorescent imaging, Autofocus hardware (Leica AFC) was used.

## Viability assay

HeLa cells stably expressing CHMP2B_GFP were seeded without treatment, with DMSO (control) or with 0.5, 1, 5, 10 or 20 μM of Bay-598. Forty-eight hours post treatment, cell viability was assessed with the cell counter Fluidlab R-300 (anvajo).

## Expression and purification of recombinant proteins

Expression and purification protocols were the same for wild type or mutant proteins. BL21 bacteria were cultured at 37 °C under agitation until reaching an OD of 0.6. Protein expression was then induced by adding 500 μM isopropyl-1-thio-β-D-galactopyranosid (IPTG) and lowering the temperature to 16 °C overnight. The bacteria were pelleted by centrifugation (4400 rpm, 20 min), washed with cold PBS, and harvested by centrifugation (4400 rpm, 15 min). Pellets were then stored at 80 °C or directly used.

In order to purify recombinant protein, bacteria were resuspended in 40 ml of lysis buffer per liter of culture (PBS 1X, 300 mM NaCl, pH 8, 1% Triton X-100, 1 mg/ml lysozyme and protease inhibitor cocktail) and incubated for 30 min at 4 °C under agitation. Lysate was then sonicated on ice (10 s ON, 20 s OFF, 7 min run and 20% power) and centrifuged (15,000 × g, 30 min, 4 °C). The supernatant was incubated 2 h on ice with either 1 ml His-select Nickel resin (Sigma) per liter of culture and 10 mM imidazole for SMYD2 and MBP-CHMP2B/CHMP3 purification or 1 ml of Glutathione-Agarose resin (Sigma) per liter of culture for GST-CHMP2B purification. Beads were then poured into a column and washed successively with washing buffer 1 (PBS 1X, 300 mM NaCl, pH 8, 0.1% Triton X-100) and washing buffer 2 (PBS 1X, 300 mM NaCl, pH 8). Proteins were eluted in elution buffer (SMYD2 and MBP-CHMP2B/CHMP3: PBS 1X, 300 mM NaCl, pH 8, 300 mM imidazole; GST-CHMP2B: 20 mM HEPES, pH 8.3, 20 mM GSH) and SMYD2 was reduced with 10 mM DTT for 20 min on ice. The purified proteins were then exchanged against storing buffer (SMYD2: 25 mM Tris-HCl, 150 mM NaCl, pH 8, 2 mM DTT; CHMP2B/CHMP3: 20 mM HEPES, pH 8.3) using a PD 10 desalting column (GE Healthcare). Protein concentration was measured using the Bradford assay with BSA as standard and protein purity was assessed by SDS-PAGE analysis. For SMYD2 protein, the 6xHis-tag was removed by SENP1 digestion. Briefly, 1 mg of purified protein was mixed with 1 μg of SENP1 SUMO-protease and incubated overnight at 4 °C under agitation. The sample was then polished over a Superdex 200 16/60 HiLoad (GE Healthcare) column in gel filtration buffer (25 mM Tris-HCl, 150 mM NaCl, pH 8, 2 mM DTT). Protein samples were aliquoted and stored at −80 °C.

## In vitro methylation assay

The lysine methyltransferase reaction was carried out overnight at 30 °C in 20 μl methylation buffer (Tris 50 mM pH 8, 50 mM NaCl, 1 mM DTT) containing 2 μg of CHMP2B (WT or K6A mutant), 2 μM of SMYD2

(WT or catalytic dead mutant Y240A) and 1 µCi of $^3$H-S-Adenosyl methionine ($^3$H-SAM, PerkinElmer). The reaction was stopped by adding 10 µL of Laemmli buffer and heating the samples at 95 °C for 5 min. Samples were separated by SDS-PAGE (NuPage 4–12%, Invitrogen) and the gel was blocked for 1 h at room temperature in blocking buffer (50% ethanol, 5% acetic acid, 5% PEG 400). Coomassie staining was used to ensure equal protein loading. The gel was then soaked in Kodak Enlightning Rapid Autoradiography Enhancer solution (PerkinElmer) to optimize the methylation signal for 1 h at room temperature, dried on a Whatman paper and exposed to a ECL Hyperfilm (Cytiva) for different time at −80 °C. The film was later developed, and methylated proteins were detected by autoradiography.

### In vitro methylation assay followed by western-blot analysis
The lysine methyltransferase reaction was carried out overnight at 30 °C in 20 µl methylation buffer (Tris 50 mM pH 8, 50 mM NaCl, 1 mM DTT) containing 2 µg of CHMP2B (WT or K6A mutant), 2 µM of SMYD2 (WT or catalytic dead mutant Y240A), 100 µM of S-Adenosyl methionine (SAM, Sigma) and 1 µM of the SMYD2 inhibitors BAY-598 or LLY-507 (Sigma). The reaction was stopped by adding 10 µl of Laemmli buffer and heating the samples at 95 °C for 5 min. Samples were separated by SDS-PAGE (NuPage 4–12%, Invitrogen) and proteins were transferred onto a nitrocellulose membrane (0.2 µm, Amersham) at 220 mA for 1 h. Ponceau staining was used to ensure equal protein loading. Membranes were blocked with non-fat milk (5%) in PBS with 1% Tween (PBST) for 1 h at room temperature and incubated with a homemade α CHMP2B-K6me1 antibody in 1% non-fat milk PBST over night at 4 °C under agitation. After washing 3 times with PBST, the membranes were incubated for 1 h at room temperature with peroxidase-coupled secondary α-rabbit antibody (SantaCruz, sc-2357). The proteins were then visualized by chemiluminescence detection using ECL reagent on LAS 4000 (Fujifilm) instrument.

### RP-UFLC-based separation and quantification of the fluorescein-labeled peptide
A 9-amino-acid peptide derived from the sequence of human CHMP2B protein and containing the lysine 6 residue was synthesized and conjugated to fluorescein amide (FAM) on its N-terminus and modified by amidation (NH2) on its C-terminus (Proteogenix). The peptide was as follows: FAM-ASLFKKKTVD-NH2. A monomethylated version of this peptide (FAM-ASLFK$_{me1}$KKTVD-NH2) was also synthesized and used as a standard control. The lysine methyltransferase reaction was carried out overnight at 30 °C in 50 µl methylation buffer (Tris 50 mM pH 8, 50 mM NaCl, 1 mM DTT) containing 75 µM of the substrate peptide, 1 µM of SMYD2, 100 µM of SAM and different concentrations of BAY-598 or LLY-507. The reaction was stopped using 50 µl of 15% perchloric acid (HClO4) (v/v) prior to injection in the instrument. Samples containing CHMP2B-K6 peptide (substrate) and its methylated form (product) were separated by RP-UFLC (Shimadzu) using Kromasil 100-5-C18 column 4.6 × 250 mm, 5 µm particle size at 40 °C. The mobile phase used for the separation consisted of two solvents. Solvent A containing water with 0.1% HClO4 and solvent B containing acetonitrile with 0.12% trifluoacetic acid (TFA). Separation was performed by an isocratic flow as followed: 79% A/21% B, rate of 1 ml/min, time of run = 30 min. CHMP2B-K6 peptide and its methylated form were monitored by the fluorescence emission ($\lambda$ = 530 nm) after excitation at $\lambda$ = 485 nm and quantified by integration of the peak absorbance area, employing a calibration curve established with various known concentrations of peptides.

### 3D modeling of CHMP2B
Homology model of CHMP2B was generated with Alphafold2 using the full-length sequence of the human CHMP2B protein as template (Q9UQN3). Display and coloring of the structure have been done with Chimera software (version 1.4). Residues have been colored depending

on their hydrophobicity according to Eisenberg scale. Red surface indicates the highest Eisenberg hydrophobicity, and blue the lowest.

### Generation of Venn diagram
The 198 reproducible SMYD2 dependent Kme1 sites from Olsen et al. 2016 study[31]—SILAC quantification of mono-methylation sites) were crossed with the flemmingsome (489 proteins found enriched at the midbody) from Addi et al. 2020 study[32] (https://flemmingsome.pasteur.cloud/). Sixteen common proteins were found and 3 of them were discriminated since they presented a SILAC ratio > 0.65, considered as non-significant Kme1 diminution. Among the remaining 13 proteins, we selected only the 3 with a known cytokinetic function.

### Anti-CHMP2B lysine 6 mono-methylation antibody purification
Anti-CHMP2B K6me1 rabbit polyclonal antibody raised against CHMP2B peptide C + ASLF-K$_{me1}$-KKTVDDV (C for Cysteine added at the N-terminus) was prepared and purified by Eurogentec (Belgium). Antiserum was obtained by immunizing rabbits with keyhole limpet hemocyanin (KLH)-conjugated peptide. The resulting IgG fraction was purified from antiserum by affinity chromatography against the CHMP2B Kme1 peptide. This antibody was used for Fig. 2A, B, C, E and Supplementary Figs. 2A, B, D, E and 5B, F. Anti-CHMP2B K6me1 rabbit polyclonal antibody raised against CHMP2B peptide (ASLF-K(Me)-KKT) was prepared and purified by Covalab. Antiserum was obtained by immunizing rabbits with (KLH)-conjugated peptide. The resulting IgG fraction was purified from antiserum by affinity chromatography against the CHMP2B K6me1 peptide. This antibody was used for Fig. 4A and Supplementary Fig. 5A.

### Samples preparation for MS analysis
For identification of CHMP2B lysine 6 methylation (K6) on endogenous CHMP2B protein, HeLa cells were lysed in lysis buffer (20 mM Tris HCl, pH 8, 150 mM NaCl, 0,4% to 1% NP40, 2 mM EDTA, protease inhibitors) and CHMP2B was immunoprecipitated (IP) with CHMP2B antibody and Dynabeads magnetic beads (Thermo Fisher Scientific). IP product was loaded on SDS PAGE gel and CHMP2B band was excised and in-gel digested by using chymotrypsine (Promega). For label Label Free Quantification (LFQ) analysis of CHMP2B lysine 6 (K6) methylation, HeLa cells were co-transfected with GFP CHMP2B and flag SMYD2 or flag SMYD2 Y240A (dead catalytic mutant). Twenty-four hours post transfection, HeLa cells were lysed in lysis buffer (20 mM Tris HCl, pH 8, 150 mM NaCl, 0.4% to 1% NP40, 2 mM EDTA, protease inhibitors) and GFP CHMP2B was IP with GFP antibody and Dynabeads magnetic beads. The IP product was loaded on SDS PAGE gel and the GFP CHMP2B band was excised and in-gel digested by using chymotrypsine. Peptide extracted from each band are vacuum concentrated to dryness and resuspended in loading buffer (0.3% TFA in miliQ water) before nanoLC-MS/MS analysis.

### LC-MS/MS analysis
Peptide extracted from each band were analyzed by coupling a RSLCnano system (Ultimate 3000, Thermo Scientific) to a Q Exactive HF-X (Thermo Scientific) mass spectrometer. Peptides were first trapped onto a C18 column (75 µm inner diameter × 2 cm; nanoViper Acclaim PepMap™ 100, Thermo Scientific) with buffer A (0.1% formic acid) at a flow rate of 2.5 µl/min over 4 min to desalt and concentrate the samples. Separation was performed on a 50 cm nanoviper column (i.d.75 µm, C18, Acclaim PepMap™ RSLC, 2 µm, 100 Å, Thermo Scientific) regulated to a temperature of 50 °C and with a linear gradient from 2% to 30% buffet B (100% acetonitrile, 0.1% formic acid) at a flow rate of 300 nl/min over 91 min. For methylation rate quantification, extracted and synthetic peptides were separated with a linear gradient from 2% to 12% over 45 min. The mass spectrometer was operated in parallel analysis monitoring (PRM) mode and MS2 scan parameters were set to select the $m/z$ ratio of CHMP2B peptides from the inclusion

lists (see acquisition list Supplementary Table 6 for characterization and Supplementary Table 7 for quantification and Supplementary Table 8 for methylation rate estimation). The MS1 scans were acquired in the $m/z$ range 375–1500 with a mass resolution of 120,000, automatic gain control (AGC) target $3 \times 10^6$, and maximum ion injection time of 50 ms. The PRM scans were acquired at a resolution of 30,000, AGC target value of $2 \times 10^5$, maximum ion injection time of 100 ms or 200 ms, isolation window of 0.7 or 1.6 $m/z$, and the normalized collisional energy (NCE) at 27 or 30.

### Data processing of MS files

For identification, the resolved raw files from the Q Exactive HF-X were searched against the GFP-CHMP2B sequence using Mascot. Enzyme specificity was set to chymotrypsin and a maximum of two-missed cleavage sites were allowed. Methionine oxidation, cysteine carbamidomethylation, N-terminal acetylation, methylation, dimethylation and trimethylation of lysine were set as variable modifications. Phosphorylation of serine and threonine were also set as variable modifications for CHMP2B characterization. Maximum allowed mass deviation was set to 10 ppm for monoisotopic precursor ions and 0.02 Da for MS/MS peaks. The resulting files were further processed using myProMS v3.9.35962[60] (https://github.com/bioinfo-pf-curie/myproms). Synthetic peptide was used to validate the presence of methylation.

For PRM quantification, Skyline was used for processing the data (version 20.1.0.155; MacCoss Lab Software, Seattle, WA; https://skyline.ms/project/home/software/Skyline/begin.view), and extracted fragment ions from each targeted masses and peak areas were integrated. Peak areas of extracted fragment ions from the same peptide ion were then summed and used as proxy for parent ion abundance.

Site abundance estimation, injection biases and changes in global variance between biological replicates ($N = 5$ for GFP_CHMP2B co-expressed with flag_SMYD2, and $N = 4$ for GFP_CHMP2B co-expressed with flag_SMYD2 Y240A, (one replicate needed to be removed for SMYD2 Y240A because not enough methylation event was detected)) were corrected by a Median and Scale normalization using a non-modified set of CHMP2B peptides. Label free quantification (LFQ) was performed at site-level following the algorithm as described[61] with the minimum number of peptide ratios set to 1 and the large ratios stabilization feature. A $t$-test was then used to evaluate the significance of the mean's difference of the log2 transformed abundances between condition SMYD2 and SMYD2 Y240A, for each site.

To estimate the methylation rate by PRM, Skyline was used for processing the data. The spectral library of the surrogate peptides was built in Skyline based on DDA MS analysis of the synthetic peptide mixture. Peak picking and peak boundaries were carried out by Skyline and manually adjusted based on the overlapping precursor and product peaks. The peak area for each transition was exported to Microsoft Excel. To build the calibration curve, the sum of the peak areas of the fragment ions of each synthetic peptide precursor charge state, 4+ and 3+, at different concentrations, as shown in Supplementary Table 9, were plotted.

### GUV in vitro test

MBP-CHMP2B 1-154, MBP-CHMP3 and SUMO-SMYD2 were used in this experiment. Methylation of CHMP2B protein was achieved by incubating 20 mg of MBP-CHMP2B with 20 mg of SUMO-SMYD2 in the presence of 1 mM SAM for 6 h at room temperature. Then, for CHMP2B (WT or methylated) and CHMP3, 6xHis MBP-tag was removed by 6xHis-tagged TEV digestion. Briefly, 20 mg of purified protein was mixed with 1 mg of TEV protease and incubated overnight at 8 °C under agitation. For CHMP3, the sample was further incubated with 1 ml of His-select Nickel resin for 2 h at 4 °C under agitation to remove TEV protease, MBP-tag and uncleaved MBP-CHMP proteins. For CHMP2B, cleavage of 6xHis MBP-tag leads to CHMP2B protein aggregation in the form of white precipitates while TEV, cleaved MBP-tag

and SUMO-SMYD2 remain in solution. Cleaved CHMP2B was then centrifuged ($15,000 \times g$, 30 min, 4 °C) and the protein pellet was washed 3 times with CHMP buffer (20 mM HEPES, pH 8.3) before being resuspended in 20 mM HEPES, pH 8.3, 6 M guanidine. To prevent protein re-aggregation, cleaved CHMP2B was serially diluted by half to reach a guanidine concentration of 375 μM, and the remaining guanidine was removed by buffer-exchange (20 mM HEPES, pH 8.3) using a PD 10 desalting column. Protein concentration was measured using absorbance measurement at 280 nm and protein purity was assessed by SDS-PAGE analysis. Methylation of CHMP2B protein was assessed by two different manners: (1) by western-blot in a similar way as for the in vitro methylation assay, (2) by mass spectrometry using a standard curve of synthetic CHMP2B K6 peptides displaying different level of methylation: CHMP2B K6me0 (KKKTVDDVIKEQNREL), CHMP2B K6me1 ($K_{me1}$KKTVDDVIKEQNREL), CHMP2B K6me2 ($K_{me2}$KKTVDDVIKEQNREL), CHMP2B K6me3 ($K_{me3}$KKTVDDVIKEQNREL). These peptides recapitulate the lysine-6-containing CHMP2B peptide generated when the protein is digested by chymotrypsin. All the protein samples were aliquoted and stored at −80 °C.

GUVs were prepared as previously described using the lipid-covered silica bead method[62]. Briefly, dioleoyil-phosphatidylcholine (DOPC), dioleoyil-phosphatidylserine (DOPS), and dioleoyl-phosphoethanolamine (DOPE), L-α-phosphatidylinositol-4,5-bisphosphate (PI(4,5)P2 and dioleoyl-phosphoethanolamine labeled with Atto 647N (Atto 647N DOPE), all purchased from Avanti Polar Lipids, were dissolved in chloroform and mixed at a molar ratio of 54.9:20:20:5:0.1%, respectively. The mixture was then dried in vacuum for 3 h in a glass vial. The dried lipid film was then hydrated in buffer solution containing 25 mM HEPES at pH 7.4 to form a suspension of multilamellar vesicles (MLVs) at 0.5 g/l. Twenty μl of MLVs were mixed with 2 μl of 40 μm silica beads (Microspheres-Nanospheres, USA), deposited on parafilm and then dried for 2 h in vacuum. The beads supporting the dried lipid films were hydrated in a 1 M trehalose solution at 60 °C and then transferred to the observation chamber with the working buffer (25 mM HEPES and 150 mM NaCl at pH 7.4). Lastly, the chamber was stirred gently for 20–30 in order to promote detachment of hydrated GUVs from the supporting silica beads. Chemical protein labeling: purified proteins were chemically labeled with alexa 568 following the instructions provided by the Alexa Fluor 568 NHS Ester Labeling Kit (Thermo Scientific, A20003 catalog number). Free labeling molecules were removed using a PD-10 Desalting Column (GE Healthcare), and the conjugate was dialyzed into the dialysis buffer containing 25 mM HEPES and 150 mM NaCl at pH 7.4. After dialysis, the conjugate was aliquoted with 10% glycerol, frozen in liquid nitrogen and stored at −80 °C. Membrane nanotube pulling from GUVs: membrane nanotubes were pulled from GUVs by direct contact between the tip of a closed glass micropipette prepared with a P-1000 micropipette puller (Sutter Instruments, USA) and the GUVs. XY micropipette position was controlled using a micro-positioning system (MP-285, Sutter Instrument, Novato, CA, USA). Desired proteins were added in the observation chamber before tube pulling at 1 μM final concentration. Quantification of sorting coefficients: CHMP3 membrane binding was quantified by measuring its integrated fluorescence and normalizing it with the integrated fluorescence of the lipid membrane. Sorting coefficients, defined as the relative change of membrane surface occupied by one protein molecule/complex, were calculated using the ratio between CHMP3 and Atto 647N DOPE integrated fluorescence on the surface of the pulled nanotube and on the GUV, neglecting the polarization factor[63], according to the following equation:

$$\text{Sorting coefficient} = \frac{(F_{\text{CHMP3}}/F_{\text{Atto 647N DOPE}})\text{nanotube}}{(F_{\text{CHMP3}}/F_{\text{Atto 647N DOPE}})\text{GUV}}$$

## HIV-1 production assay

For a HIV-1 production assay, HeLa stably expressing GFP, CHMP2B_GFP, CHMP2B K6R_GFP or CHMP2B K6A_GFP were transfected with HIV-1 proviral DNA NL4-3. Transfections of HeLa cells with NL4-3 HIV-1 proviral DNA were performed using Lipofectamine LTX with PLUS Reagent (Life technologies), following the manufacturer's instructions.

In a single round of infection, HeLa cells were treated with siRNA (25 nM) using Lipofectamine RNAiMAX (Life TechnologiesSpecific). Forty-eight hours after, siRNA-treated Hela cells were transfected with HIV-1 proviral DNA NL4-3 for 4 h in OPTIMEM.

Forty-eight hours after proviral transfection, supernatants were then collected, centrifuged 5 min at $500 \times g$, 0.45 μm-filtered and used for HIV-1 CAp24 quantification by ELISA (released CAp24) (Perkin Elmer). Viral particles released into the supernatant were pelleted through a 20% sucrose cushion by ultracentrifugation at $150,000 \times g$ for 60 min and resuspended in laemmli sample buffer. Equal volumes of pelleted viruses were analyzed by western blotting using mouse anti-CAp24. Cell lysates were analyzed by western blotting.

## HIV-1 infectivity assay

In a single round infectivity assay, the titers of released viruses were determined by infection of the indicator cells HeLa P4R5 in a standardized 96-well titration assay by luminometric analysis of β-galactosidase activity (Kit Galacto-*Star*™ β-Galactosidase Reporter Gene Assay System, Life Technology) following the manufacturer's instructions.

## Statistical analysis

The statistical details of all experiments are reported in the figure including statistical analysis performed, error bars, statistical significance and exact $N$ numbers. Statistical tests used in this study were Two-sided. Data normality was assessed using the Shapiro–Wilk test. In the case of multiple sample comparisons, we employed the "two-stage" Benjamini, Krieger, and Yekutieli procedure to control the false discovery rate (FDR). Statistics were performed using GraphPad Prism 6 or 9 software.

## Reporting summary

Further information on research design is available in the Nature Portfolio Reporting Summary linked to this article.

## Data availability

All data are available in the main text, the Supplementary Materials and the Source data[64]. The mass spectrometry proteomics data have been deposited to the ProteomeXchange Consortium (http://proteomecentral.proteomexchange.org) via the PRIDE partner repository[65] with the dataset identifier PXD041760 (https://ftp.pride.ebi.ac.uk/pride/data/archive/2024/03/). Source data are provided with this paper.

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

## Acknowledgements

We thank all members of the Weitzman laboratory for critical reading of the manuscript and for helpful discussions. We thank members of the UMR7216 and Winfried Weissenhorn for helpful discussions. We are grateful to the support of the technical platforms at the Université Paris Cité including the EPI², the Vectorology and the BiBs Platforms, all hosted in UMR7216 Epigenetic and Cell Fate, for technical advice and access to instruments. We thank the ImagoSeine core facility of the Institut Jacques Monod, member of the France BioImaging infrastructure (ANR-10-INBS-04). SM acknowledges fundings from the CNRS, the IdEx Université Paris Cité, ANR-18-IDEX-0001-SM, by the "Who Am I?" Laboratory of Excellence #ANR-11-LABX-0071-SM funded by the French Government through its "Investments for the Future" program operated by the ANR under grant #ANR-11-IDEX-0005-01-SM, by the Fondation ARC pour la Recherche sur le Cancer (ARC no. 155029-SM), la LIGUE (No. 207104, RS20/75-34-SM). S.M. was a junior member of the Institut Universitaire de France (IUF 2012ND 3369-SM). The JBW team is supported by the EUR G.E.N.E. (#ANR-17-EURE-0013-JW) and is a labeled team of the Fondation pour la Recherche Médicale (Equipe FRM #EQU202203014701-JW). A.R. was a recipient of a MESRI PhD fellowship from the French government (Doctoral School BioSPc) and a 4th year PhD fellowship from the Fondation pour la Recherche Médicale (FDT202204014904-AR). D.J. holds a fellowship from ANRS (ECTZ60924-DJ) and then from SIDACTION (2021-

2-FJC-13113-DJ). J.E. acknowledges support from EMBO long-term fellowship ALTF 989-2022-JE. A.R. acknowledges funding from the Swiss National Fund for Research, grants No. 310030_200793/1-AR and No. CRSII5_189996-AR, and from the European Research Council, Synergy Grant No. 951324_R2-TENSION-AR.

## Author contributions

A.R., J.B., T.A., G.J., A.A., A.C., and S.M. performed experiments and analyzed the results related to cytokinesis. D.J. and T.A. contributed equally to this work. D.J. and C.B.T. performed experiments and analyzed the results related to virus budding experiments. J.B., J.E and A. Roux performed experiments and analyzed the results related to in vitro reconstitution membrane. D.L. and B.L. performed and analyzed the Mass-spectrometry experiments. T.A. taught time-lapse microscopy analysis to A.R. J.B.W., A.E., C.B.T., and N.R. provided reagents and helpful discussions. A.E. provided constructive discussions and invaluable advice for this study. A.R., J.B., and S.M. designed the study, performed experiments, analyzed the results. A.R. and S.M. wrote the manuscript. J.B.W., A.E., and C.B.T. reviewed and edited the manuscript. S.M. developed the concept, provided overall supervision and secured funding.

## Competing interests

The authors declare no competing interests.
