## [Peer Review File · Nature Communications]

REVIEWER COMMENTS

Reviewer #1 (Remarks to the Author):

Jérémy Berthelet and co-workers propose in their manuscript that 'Methylation of ESCRT-III components regulates the timing of cytokinetic abscission'. In particular, the data suggests that the lysine methyl transferase SMYD2 localizes to the midbody during abscission, where it methylates the ESCRT-III subunit Chmp2B on its n-terminal lysine residue 6 (K6). This methylation event contributes to the timing of cytokinesis and HIV budding. The manuscript is interesting because it implicates for the first time methylation as a mode to control ESCRT mediated membrane remodeling event.

While it is relatively clear that Smyd2 can methylate Chm2B on lysine 6, I'm not convinced that this methylation event controls the timing of cytokinesis or HIV budding. My main arguments are as follows and these major points should be addressed prior to publication:

(1) Inhibition of Smyd2 function (using inhibitor or by siRNA mediated knock down) caused only a minimal delay of cytokinetic abscission (Fig.4 B, D), while the expression of Chmp2B-K6A caused a much stronger delay in cytokinesis and phenocopied loss of Chmp2B function (Fig. 3C, D). This is also true in a setting, where Chmp2B-K6A is expressed and Smyd2 function is inhibited (Fig.4J). These findings also apply for HIV release (Fig 6).

Hence, based on the data, it is clear that Chmp2B-K6A acts like a loss of function mutant, while the inhibition of the lysine methyl transferase has milder effects.

Hence it seems to me that the effects of the Chmp2B-K6A mutant on cytokinesis or HIV budding are not caused by lack of methylation, but perhaps by the lack of a positive charge at this position. This is not entirely unlikely, given the n-terminal lysine residues of ESCRT-III proteins are often used for membrane interaction or for the interaction with other ESCRT-III subunits. I'm wondering if a Chmp2B-K6R mutant would be better suited to dissect the role of lysine methylation.

(2) It remains mechanistically unclear how the methylation would control Chmp2B function. Does methylation affect Chmp2B activation, membrane interaction, or the interaction with other ESCRT proteins?

(3) The immunofluorescence experiment in Fig 2B,C should be conducted with cells expressing Chmp2B-K6R and Chmp2B-K6A.

Minor points:

(1) The authors make a point that their methylation specific Chmp2B does not recognize the endogenous CHMP2B_K6me1 by western blot analysis of protein extracts. Have tried to immune-precipitate the endogenous CHMP2B and then use the CHMP2B_K6me1 antibody?

(2) Rather than using a monomeric Chmp2B model (Figure S2E), it would perhaps be more informative to model Chmp2B in an ESCRT-III the polymer based on Chmp2A/Chmp3 structure – methylation (or just mutation of the the positive charge at K6) might change membrane binding properties as well as interaction with Chmp3

(3) To estimate the level of Chmp2B-GFP and Chmp2B-k6A-GFP levels relative to the levels of endogenous Chmp2B, the author should show protein levels on the same WB blot using a Chmp2B antibody.

Reviewer #2 (Remarks to the Author):

The Richard et al “Methylation of ESCRT-III components regulates the timing of cytokinetic abscission” manuscript identifies CHMP2B as a new substrate of SMYD2. Specifically, SMYD2 mono-methylation of CHMP2B at lysine 6 is proposed to occur at the ICB and to promote abscission during cytokinesis. The authors also provide evidence that in a manner dependent on CHMP2B methylation, the overexpression of SMYD2, which is frequently seen in cancer cells, promotes abscission checkpoint bypass to overcome cytokinetic challenge. The manuscript also shows a role for CHMP2B methylation in HIV-1 budding.

Overall, the manuscript provides some of the first insight into the role of protein methylation signaling in ESCRT functions and provides new clarity into the function of SMYD2. Therefore, I recommend for publication.

Reviewer #3 (Remarks to the Author):

The manuscript „Methylation of ESCRT-III components regulates the timing of cytokinetic abscission” by Richard et al. presents the study of a methylation mechanism regulating the dynamics of abscission in the final stages of cell division. The authors demonstrate that the methylating enzyme SMYD2 modifies a lysine in CHMP2B, a component of ECRT-II complex, and in this way tunes the abscission process. The experimental design of the project is well rounded with many orthogonal approaches and the work offers the explanation of a fundamental biological mechanism, so it will be of interest to a broad readership. There are a few concerns to be addressed before publication.

Detailed comments:

1. Page 5, lines 56-63. How did the authors make sure that the antibody they used recognizes methylation per se? Why did it not recognize the endogenous protein in the Western blot?
2. Page 7, lines 18-20. Why only partially? This is counterintuitive, the amplitude with the inactive enzyme should be the same as for unmethylable mutant, is there a residual activity left or is there another enzyme that could modify the same lysine in CHMP2B? If siRNA leaves any residual activity (show?), wouldn't CRISPR be a better method?
3. How are the experiments in HeLa cells (page 8) relevant to cancer?
4. Page 17, line 50 (and related figures) how was the 1 μ M concentration of both inhibitors determined as the correct one in this system? Have the authors performed titration? How about cell death when the inhibitors are applied? It is concerning that the difference between control and BAY-598 treated samples (Figure 2) is so small, it may be because the inhibitor does is not optimized. Also, there are obvious outliers that weren't removed and probably are skewing the statistical analysis.

We thank all three referees for their diligent review of our manuscript, for their constructive comments, and thoughtful suggestions. We were pleased to note that all three reviewers agree on the novelty of our study, highlighting its appeal to a wide readership. We have thoroughly addressed all of their questions and concerns in the revised manuscript, as outlined below.

REVIEWER COMMENTS

Reviewer #1 (Remarks to the Author):

Jérémy Berthelet and co-workers propose in their manuscript that ‘Methylation of ESCRT-III components regulates the timing of cytokinetic abscission’. In particular, the data suggests that the lysine methyl transferase SMYD2 localizes to the midbody during abscission, where it methylates the ESCRT-III subunit Chmp2B on its n-terminal lysine residue 6 (K6). This methylation event contributes to the timing of cytokinesis and HIV budding. The manuscript is interesting because it implicates for the first time methylation as a mode to control ESCRT mediated membrane remodeling event.

Response : We thank Reviewer 1 for this positive feedback and for highlighting the novelty of our study. We have conducted all requested experiments as detailed below.

While it is relatively clear that Smyd2 can methylate Chm2B on lysine 6, I’m not convinced that this methylation event controls the timing of cytokinesis or HIV budding. My main arguments are as follows and these major points should be addressed prior to publication: (1) Inhibition of Smyd2 function (using inhibitor or by siRNA mediated knock down) caused only a minimal delay of cytokinetic abscission (Fig.4 B, D), while the expression of Chmp2B-K6A caused a much stronger delay in cytokinesis and phenocopied loss of Chmp2B function (Fig. 3C, D). This is also true in a setting, where Chmp2B-K6A is expressed and Smyd2 function is inhibited (Fig.4J). These findings also apply for HIV release (Fig 6). Hence, based on the data, it is clear that Chmp2B-K6A acts like a loss of function mutant, while the inhibition of the lysine methyl transferase has milder effects.

Response : We thank Reviewer 1 for the insightful comments. We address these concerns below:

To clarify the amplitude differences between CHMP2B-K6A mutant and decreased K6 methylation, we added a sentence to the revised manuscript (line 223): "The difference in amplitude observed between the use of CHMP2B K6 mutants and SYMD2 depletion/inhibition suggests that mutating the K6 residue has a stronger physicochemical impact than inhibiting its methylation."

Role of CHMP2B methylation in cytokinetic abscission and HIV-1 budding : We should emphasize that our conclusion regarding the role of SMYD2-mediated CHMP2B methylation in regulating the timing of cytokinesis is supported by both loss-of-function (LOF) and gain-of-function (GOF) data. The degree of delay in abscission with SMYD2 LOF is moderate but reproducible and statistically significant as indicated by the p-values ($p=0.0078$ and $p<0.0001$ for genetic and pharmacological SMYD2 LOF respectively; Fig. 4 B, D), and in HIV-1 release

($p=0.028$; Fig. 6E) demonstrating the involvement of SMYD2. It is worth noting that SMYD2 depletion or inhibition likely leads to a significant reduction in CHMP2B methylation rather than a complete loss. Methylated CHMP2B molecules might still be present at the midbody (Fig. 2C-E), either due to residual SMYD2 activity or the possibility of another methyltransferase contributing to residual CHMP2B methylation.

Importantly, abscission delay upon SMYD2 LOF is only observed in a CHMP2B WT genetic context and is not observed when using the CHMP2B K6A mutant (Fig. 4H-I) indicating that the role of SMYD2 in abscission is mediated by the K6 residue of CHMP2B.

Our conclusion that methylation regulates the timing of cytokinesis are further supported by SMYD2 GOF experiments, that showed an acceleration of abscission timing which is dependent on SMYD2 catalytic activity (Fig. S6C-F). SMYD2 GOF bypassed the abscission checkpoint and leads to a premature abscission, by promoting CHMP2B relocation at the abscission site (Fig. 5C-G)

Several observations including, (i) the abscission delay observed with SMYD2 LOF in a CHMP2B WT background, (ii) the absence of delay in a CHMP2B K6A mutant background, (iii) and the acceleration of abscission with SMYD2 GOF in cells undergoing cytokinesis, collectively support our hypothesis that lysine methylation signaling fine-tunes the ESCRT-III machinery to regulate the timing of cytokinetic abscission.

We hope these clarifications provide a more comprehensive understanding of our study.

Hence it seems to me that the effects of the Chmp2B-K6A mutant on cytokinesis or HIV budding are not caused by lack of methylation, but perhaps by the lack of a positive charge at this position. This is not entirely unlikely, given the n-terminal lysine residues of ESCRT-III proteins are often used for membrane interaction or for the interaction with other ESCRT-III subunits. I'm wondering if a Chmp2B-K6R mutant would be better suited to dissect the role of lysine methylation.

Response: The reviewer indeed is right to raise the issue of charge. We agree that the CHMP2B K6A mutant may through the loss of a positive charge, impeding anchoring to the membrane or interactions with other ESCRT-III subunits. As suggested by the reviewer, we have extended our investigation and established a stable HeLa cell line expressing the CHMP2B K6R mutant. Using time-lapse spinning-disk confocal microscopy, we found that mutation of lysine 6 to arginine resulted in a delayed abscission, with a slower recruitment of CHMP2B K6R at the midbody and reduced progression to the abscission site compared to CHMP2B WT (we have included these data in the new Fig. S3F-H). In addition, cells expressing the CHMP2B K6R mutant exhibited a reduction in the release of HIV-1 particles and inhibited the production of infectious virions (these results are shown in the new Fig. 6A-C). Thus, while the residue charge remains unchanged, the un-methylatable mutant CHMP2B K6R delays abscission and reduces HIV-1 release.

Altogether, these additional findings reinforce the significance of CHMP2B lysine 6 in scission timing, influencing CHMP2B transition to the abscission site during cytokinesis and in HIV-1 release. We added these new data to the revised manuscript (Fig. 6A-C and Fig. S3F-H), and commented on these results in the revised text (p.7, line 208-213 and p.10, line 306-308). We appreciate the opportunity to address this concern and feel that the new results further strengthen the conclusion of our work.

(2) It remains mechanistically unclear how the methylation would control Chmp2B function. Does methylation affect Chmp2B activation, membrane interaction, or the interaction with other ESCRT proteins?

Response: The reviewer raises a pertinent question regarding the mechanistic underpinnings of how methylation influences CHMP2B function. To investigate the potential mechanism(s) associated with the *in cellulo* effect observed with the methylated CHMP2B, we performed *in vitro* reconstitution experiments with purified ESCRT-III proteins and model membranes. We used C-terminally truncated versions of CHMP2B which facilitates activation of the ESCRT-III proteins. We optimized the conditions to achieve efficient methylation of CHMP2B K6, representing 70% of the total recombinant CHMP2B protein *in vitro*. This enabled us to explore whether methylation could modulate ESCRT-III membrane binding and polymerization properties. We generated membrane nanotubes pulled from Giant Unilamellar Vesicles (GUVs) to study the effect of CHMP2B methylation on CHMP3 recruitment and polymerization. Our findings indicate that methylation of CHMP2B promotes the formation of ESCRT-III filaments on curved membrane (we have added these new data to new Fig. 4J-K). This finding provides mechanistic insight into how post-translational modifications can regulate the functions of ESCRT-III at the molecular level, ultimately influencing the timing and efficiency of the scission process. To perform these new experiments we have established a collaboration with experts in the field; we subsequently added two new authors to the manuscript (J. espadas and A.Roux).

These *in vitro* data are now included in new Fig. 4J-K and revised text p. 8 (line 240-261) and p. 11 (line 372-377).

(3) The immunofluorescence experiment in Fig2B,C should be conducted with cells expressing Chmp2B-K6R and Chmp2B-K6A.

Response: We conducted additional experiments in response to the reviewer's suggestion. The experiments have been extended to include cells expressing both CHMP2B K6R and CHMP2B K6A proteins. Our experiments using HeLa cell lines expressing WT CHMP2B, CHMP2B K6A, and CHMP2B K6R show that while SMYD2 expression increased WT CHMP2B methylation at the intercellular bridge of dividing cells, this effect was not observed when lysine 6 of CHMP2B was mutated either to alanine (A) or arginine (R). These new findings underscore the specificity of recognition of SMYD2-mediated CHMP2B K6 methylation in cells. These data have been incorporated into new Fig. S2E, and commented on in the revised manuscript (p. 6, line 170-172).

Minor points:

(1) The authors make a point that their methylation specific Chmp2B does not recognize the endogenous CHMP2B_K6me1 by western blot analysis of protein extracts. Have tried to immune-precipitate the endogenous CHMP2B and then use the CHMP2B_K6me1 antibody?

Response : Using immunoprecipitation of endogenous CHMP2B, we have now successfully detected CHMP2B K6me1 (see new data added to new Fig. 2B). Importantly, the endogenous

CHMP2B methylation signal was lost upon treatment with a pharmacological inhibitor of SMYD2 (new Fig. 2B, revised text p5, line 162-163).

(2) Rather than using a monomeric Chmp2B model (Figure S2E), it would perhaps be more informative to model Chmp2B in an ESCRT-III the polymer based on Chmp2A/Chmp3 structure – methylation (or just mutation of the positive charge at K6) might change membrane binding properties as well as interaction with Chmp3

Response : We appreciate the insightful suggestion from the reviewer concerning the modeling of CHMP2B in an ESCRT-III polymer based on the CHMP2A /CHMP3 structure. However, it is important to note that the structures of the very N-terminal part of CHMP2B and CHMP2A have not been experimentally resolved. Alpha Fold predicted that this region is disordered, making it challenging to model how the mutation of K6 might impact the CHMP2B/CHMP3 polymer structure. To address this limitation, we conducted *in vitro* experiments to explore the functional interaction between methylated CHMP2B and CHMP3, as described in the answer to comment 2 above. Our findings suggest that CHMP2B K6 methylation facilitates the formation of ESCRT-III filaments on curved membrane (new Fig. 4J-K).

(3) To estimate the level of Chmp2B-GFP and Chmp2B-k6A-GFP levels relative to the levels of endogenous Chmp2B, the author should show protein levels on the same WB blot using a Chmp2B antibody.

Response : We agree that this information is required, and we clarify that this information was indeed presented in the initial manuscript, specifically in Supplementary Fig. S3B-D.

Reviewer #2 (Remarks to the Author):

The Richard et al “Methylation of ESCRT-III components regulates the timing of cytokinetic abscission” manuscript identifies CHMP2B as a new substrate of SMYD2. Specifically, SMYD2 mono-methylation of CHMP2B at lysine 6 is proposed to occur at the ICB and to promote abscission during cytokinesis. The authors also provide evidence that in a manner dependent on CHMP2B methylation, the overexpression of SMYD2, which is frequently seen in cancer cells, promotes abscission checkpoint bypass to overcome cytokinetic challenge. The manuscript also shows a role for CHMP2B methylation in HIV-1 budding.

Overall, the manuscript provides some of the first insight into the role of protein methylation signaling in ESCRT functions and provides new clarity into the function of SMYD2. Therefore, I recommend for publication.

Response : We sincerely appreciate the positive feedback from Reviewer 2 and his/her recognition of the novel insights our manuscript offers into the role of protein methylation signaling in ESCRT functions, shedding light on the function of SMYD2. We are grateful for the recommendation for publication in *Nature Communications*.

Reviewer #3 (Remarks to the Author):

The manuscript „Methylation of ESCRT-III components regulates the timing of cytokinetic abscission” by Richard et al. presents the study of a methylation mechanism regulating the dynamics of abscission in the final stages of cell division. The authors demonstrate that the methylating enzyme SMYD2 modifies a lysine in CHMP2B, a component of ECRT-II complex, and in this way tunes the abscission process. The experimental design of the project is well rounded with many orthogonal approaches and the work offers the explanation of a fundamental biological mechanism, so it will be of interest to a broad readership. There are a few concerns to be addressed before publication.

Response : We thank Reviewer 3 for his/her constructive feedback. We sincerely appreciate the positive remarks from Reviewer 3 regarding the broad appeal of our work and the robustness of the experimental design. We have made the necessary revisions to address the Reviewer’s concerns.

Detailed comments:

1. Page 5, lines 56–63. How did the authors make sure that the antibody they used recognizes methylation per se?

Response : To ensure the specificity of the anti-CHMP2B K6me1 antibody for recognizing methylation, we conducted a series of experiments, and the following lines of evidence strongly support its selectivity for CHMP2B K6 methylation:

Antibody’s specificity for CHMP2B

We demonstrated that the anti-CHMP2B K6me1 antibody signal was reduced in experiments using CHMP2B siRNA, supporting the antibody’s specificity for CHMP2B (Fig. S2D).

Antibody’s specificity for CHMP2B lysine 6

Mutating the K6 residue prevents the recognition by the anti-CHMP2B K6me1 antibody. The antibody did not detect the K6me1 signal in protein extracts from cells expressing the CHMP2B K6A mutant, confirming its high specificity for the lysine 6 of CHMP2B (Fig. 2A; Fig. S2A).

Antibody’s specificity for CHMP2B lysine 6 methylation

The recognition by the anti-CHMP2B K6me1 antibody is dependent on SMYD2 catalytic activity. We demonstrated that the anti-CHMP2B K6me1 antibody successfully detected CHMP2B in the presence of SMYD2, but not in the presence of the catalytic-dead SMYD2 Y240A mutant enzyme (Fig. 2A; Fig. S2A). The methylation signal was lost upon either SMYD2 silencing or SMYD2 pharmacological inhibition, providing additional confirmation of the antibody’s specificity (Fig. 2C-D; Fig. S5B; Fig. S5F). Collectively, these results demonstrate that the antibody recognizes SMYD2-dependent methylation events.

In conclusion, the collective evidence, including biochemical assays, mutant studies, recombinant protein validations, and immunofluorescence analysis, supports the conclusion

that the anti-CHMP2B K6me1 antibody specifically recognizes CHMP2B K6 methylation *in vitro* and in cells.

Why did it not recognize the endogenous protein in the Western blot?

We appreciate the reviewer's insightful question. Using immunoprecipitation of endogenous CHMP2B, we have now successfully detected CHMP2B K6me1 (new Fig. 2B). Importantly, the endogenous CHMP2B methylation signal was lost upon treatment with a pharmacological inhibitor of SMYD2 (new Fig. 2B). We appreciate the opportunity to address this point which provides valuable insights into the methylation status of endogenous CHMP2B.

2. Page 7, lines 18-20. Why only partially? This is counterintuitive, the amplitude with the inactive enzyme should be the same as for unmethylable mutant, is there a residual activity left or is there another enzyme that could modify the same lysine in CHMP2B? If siRNA leaves any residual activity (show?)the siRNA, wouldn't CRISPR be a better method ?

To clarify the amplitude differences between CHMP2B K6A mutant and decreased K6 methylation, we added a sentence to the revised manuscript (line 223): "The difference in amplitude observed between the use of CHMP2B K6 mutants and SYMD2 depletion/inhibition suggests that mutating the K6 residue has a stronger physicochemical impact than inhibiting its methylation."

We hope this additional information addresses Reviewer 3's concerns and enhances the clarity of our findings.

3. How are the experiments in HeLa cells (page 8) relevant to cancer?

HeLa cells were selected as a model system due to their wide use in studying abscission and the abscission checkpoint, providing a well-established context for our investigations. Furthermore, to strengthen the relevance of our findings, we extended our experiments to include U2OS osteosarcoma cells. These results in Fig. S4A-B, show that SMYD2 is also required for CHMP2B proper localization at the intercellular bridge (ICB) in different cell types undergoing cytokinesis and illustrate a conserved requirement for SMYD2 in governing the dynamics of abscission across distinct cell types.

4. Page 17, line 50 (and related figures) how was the 1 uM concentration of both inhibitors determined as the correct one in this system? Have the authors performed titration? How about cell death when the inhibitors are applied?

To address Reviewer 3 questions, we conducted a thorough titration of BAY-598 across concentrations (0.5, 1, 5, 10, and 20 μ M). Importantly, for these titration experiments, we did not observe any cell death associated with the use of BAY-598 (new Fig. S2B). The *in vitro* experiments revealed a maximal inhibitory effect on CHMP2B methylation within the BAY-598 concentration range of 4 to 20 μ M. Considering the balance between achieving an effective inhibitory impact on CHMP2B methylation and minimizing any potential adverse effects, we selected a concentration of 10 μ M for our cell-based assays. At this concentration, we observed a loss of CHMP2B methylation without any associated-cell death.

This information clarifies the rationale behind the chosen concentration.

It is concerning that the difference between control and BAY-598 treated samples (Figure 2) is so small, it may be because the inhibitor does is not optimized. Also, there are obvious outliers that weren't removed and probably are skewing the statistical analysis.

The concentration of BAY-598 used is 10 μ M which decreases the methylation of CHMP2B in protein extract (new Fig. S2B).

The Mann-Whitney test performed in the paper compares ranks instead of actual data point so it is a quite robust test in case of datasets including outliers.

As suggested by the Reviewer, we conducted a thorough statistical analysis by removing the outliers: the updated results are shown here for the Reviewer and confirm the statistical significance ($p < 0.0001$).

REVIEWERS' COMMENTS

Reviewer #1 (Remarks to the Author):

The authors have addressed our major points during the revision. The new data support the concept that methylation of Chmp2b regulates ESCRT-III function to time cytokinetic abscission. This is a novel and interesting finding and should be published.

Reviewer #2 (Remarks to the Author):

The authors have addressed my previous concerns and I recommend publication.

Reviewer #3 (Remarks to the Author):

The authors have addressed my concerns.